
# Large difference in aerosol radiative effects from BVOC-SOA treatment in three ESMs

Moa K. Sporre[1,2], Sara M. Blichner[1], Roland Schrödner[3], Inger H. H. Karset[1], Terje K. Berntsen[1,4], Twan van Noije[5], Tommi Bergman[5,6], Declan O'Donnell[6], and Risto Makkonen[6,7]

[1]Department of Geosciences, University of Oslo, Postboks 1022 Blindern, 0315 Oslo, Norway
[2]Now at: Department of Physics, Lund University, Box 118, 22100 Lund, Sweden
[2]Institute for Tropospheric Research, Permoserstr. 15, 04318 Leipzig, Germany
[4]CICERO Center for International Climate Research, Postboks 1129 Blindern, 0318 Oslo, Norway
[5]Royal Netherlands Meteorological Institute (KNMI), PO Box 201, 3730 AE De Bilt, the Netherlands
[6]Climate System Research, Finnish Meteorological Institute, P.O. Box 503, FI-00101, Helsinki, Finland
[7]Institute for Atmospheric and Earth System Research / Physics, Faculty of Science, University of Helsinki, P.O. Box 64, FI-00014, Finland

**Correspondence:** Moa Sporre (moa.sporre@nuclear.lu.se)

**Abstract.** Biogenic volatile organic compounds (BVOCs) emitted from vegetation are oxidized in the atmosphere and can form aerosol particles either by contributing to new particle formation or by condensing onto existing aerosol particles. As the understanding of the importance of BVOCs for aerosol formation has increased over the past 10 years these processes have made their way into Earth System Models (ESMs). In this study, sensitivity experiments are run with three different ESMs, (the

Norwegian Earth System Model (NorESM), EC-Earth and ECHAM) to investigate how the direct and indirect aerosol radiative effects are affected by changes in the formation of secondary organic aerosol (SOA) from BVOCs. In the first two sensitivity model experiments, the yields of SOA precursors from oxidation of BVOCs are changed by ±50%. For the third sensitivity test, the formed oxidation products do not participate in the formation of new particles, but are only allowed to condense onto existing aerosols. In the last two sensitivity experiments, the emissions of BVOC compounds (isoprene and monoterpenes) are

turned off, one at a time.

The results show that the impact on the direct radiative effect (DRE) are linked to the changes in the SOA production in the models, where more SOA leads to a stronger DRE and vice versa. The magnitude by which the DRE changes (maximally 0.15 Wm$^{-2}$ globally averaged) in response to the SOA changes however varies between the models, with EC-Earth displaying the largest changes. The results for the cloud radiative effects (CRE) are more complicated than for the DRE. The changes

in CRE differ more among the ESMs and for some sensitivity experiments they even have different signs. The most sensitive models are NorESM and EC-Earth, which has CRE changes of up to 0.82 Wm$^{-2}$. The varying responses in the different models are connected to where in the aerosol size distributions the changes in mass and number due to SOA formation occur, in combination with the aerosol number concentration levels in the models. We also find that interactive gas-phase chemistry as well as the new particle formation parameterization have important implications for the DRE and CRE in some of the

sensitivity experiments. The results from this study indicate that BVOC-SOA treatment in ESMs can have a substantial impact on the modelled climate but that the sensitivity varies greatly between the models. Since BVOC emissions have changed





historically and will continue to change in the future, the spread in model results found in this study introduces uncertainty into ESM estimates of aerosol forcing from land-use change and BVOC feedback strengths.

## 1 Introduction

The climatic relevance of biogenic volatile organic compounds (BVOCs) emitted from vegetation has received increasing attention over the past years. Emitted BVOCs are quickly oxidized in the atmosphere producing a number of different products with lower volatility. These can then form secondary organic aerosols (SOA), increasing both aerosol number concentration (through new particle formation (NPF), and participation in early growth) and aerosol sizes (through condensation onto pre-existing particles) (Shrivastava et al., 2017). The formation of SOA from BVOCs can thus influence climate both through

changes in cloud properties (indirect aerosol effects) (Twomey, 1974; Albrecht, 1989) and through changes in scattering and absorption of solar radiation by aerosols (direct aerosol effect).

BVOC emissions depend on various environmental factors, in particular temperature, $CO_2$ concentrations and land use and are thus expected to have changed in the past and to continue to change in the future (e.g. Bonan, 2016, sec. 31.6). Studies have found that future BVOC emissions are likely to increase due to warming and higher $CO_2$ concentrations, and that BVOCs

could dampen temperature increase and provide a negative climate feedback (Sporre et al., 2019; Paasonen et al., 2013; Kulmala et al., 2014; Scott et al., 2018; Carslaw et al., 2010). It is thus important to include these processes in Earth System Models (ESMs) to estimate aerosol effects in the future, but also in the past. SOA formation has been added to many models over the recent years in response to the increased understanding of the importance of BVOCs to aerosol formation. However, uncertainties regarding these processes in models are large, e.g. Tsigaridis et al. (2014) show an order of magnitude variation

between the 31 models in the vertical profile of organic aerosol mass in their intercomparison.

Organics constitute a large fraction of the atmospheric aerosol mass (Shrivastava et al., 2017; Zhang et al., 2007) and as much as 50-85 % of this can be SOA (Zhang et al., 2007; Glasius and Goldstein, 2016). In model estimates, biogenic SOA usually dominates the SOA budget (Glasius and Goldstein, 2016; Hallquist et al., 2009; Kelly et al., 2018). As an exception, Shrivastava et al. (2015) find biomass and fossil fuel burning to be the largest source, and biogenics to be the second largest. Moreover,

Spracklen et al. (2011) claim the biogenic SOA formation is anthropogenically controlled.

The SOA formation and processing pathways in the atmosphere are remarkably complex. To represent these in ESMs, a trade-off must be made between detail and computational cost (Tsigaridis et al., 2014). In reality, BVOCs consist of a myriad of compounds with different properties. However, in ESMs these are often represented by 2-3 tracers, usually isoprene, monoterpenes (MTs) and sesquiterpenes which are estimated to constitute around 50%, 15% and 3% respectively of the total BVOC

emissions (Guenther et al., 2012). The oxidation products of BVOCs, while in reality a large variety of compounds produced through a series of reactions (Glasius and Goldstein, 2016; Shrivastava et al., 2017), are lumped into a few tracers which can condense onto existing aerosols or contribute to NPF and early growth (Tsigaridis et al., 2014). Not all oxidation products have low enough volatility to be relevant for aerosol formation. Therefore, the percentage of low volatility products formed during the oxidation is described by yields for each oxidation reaction (e.g. Tsigaridis et al., 2014; Jokinen et al., 2015; Makkonen





et al., 2014). Some models also use volatility basis sets or similar approaches to account for changes in volatility during the oxidation (Donahue et al., 2006, 2011; Yu, 2011).

All three models included in this study use two tracers representing the oxidation products from the BVOCs. One tracer represents the highly oxidized BVOCs which can take part in NPF and the early growth of the newly formed particles. This tracer will be denoted ELVOCs (Extremely Low-Volatile Organic Compounds) here. The other tracer represents the oxidation

products with somewhat higher volatility that can condense onto larger aerosols, and will be denoted by L/SVOCs (Low/Semi-Volatile Organic Compounds). The VOCs are however not actually volatile in these models since the parameterizations only allow irreversible condensation of the organics.

As mentioned, evidence suggests that low-volatility organics contribute at the earliest stages of NPF (Tröstl et al., 2016; Riccobono et al., 2014; Ehn et al., 2014; Kirkby et al., 2016; Riipinen et al., 2011, 2012; Zhang et al., 2012b) and this is

increasingly considered in global models. The formation / nucleation rate of new particles is typically parameterized with one parameterization of binary nucleation of sulfuric acid ($H_2SO_4$) and water ($H_2O$) vapours for the entire atmosphere. However, since these nucleation parameterizations underestimate NPF in the boundary layer (BL) (Spracklen et al., 2006), an additional parameterization involving sulphuric acid and organics that better capture the NPF in the BL is often added to the ESMs. These parameterizations are not always limited to the BL but in this manuscript we will refer to them as BL nucleation since they

were introduced into the models to address the underestimation of NPF there.

There are large uncertainties in several of the processes representing SOA formation in ESMs. BVOC emissions are poorly constrained both locally and globally (Heald and Spracklen, 2015) and future changes in emissions are highly uncertain, both because of scenario uncertainty and because vegetation response is uncertain (Hantson et al., 2017). In addition, the formation of low-volatility oxidation products depends on a number of variables, including oxidation capacity, $NO_x$ concentrations (Shri-

vastava et al., 2017), specific BVOC species etc., which results in large uncertainties in the yields (Jokinen et al., 2015). The representation of SOA in global models is currently under rapid development (Tsigaridis et al., 2014; Makkonen et al., 2014; Gordon et al., 2016, 2017; Makkonen et al., 2012; Dunne et al., 2016). It is important to understand the dynamics introduced by these parameterizations and how they interact with the other parts of the models – in particular related to the direct and indirect aerosol effect, which have strong impacts on climate.

In this study we investigate the impact of choices in emissions of SOA precursors and yields of BVOC oxidation products on the climatic effects of SOA through a series of sensitivity experiments with three ESMs. The models have comparable treatments of SOA formation but have different aerosol schemes and different treatment of gas-phase chemistry. The comparison of the simulations for the different models and experiments thus gives us the possibility to investigate the sensitivities to common parameters. We investigate how the direct and indirect aerosol effects are impacted by the changes in yields and emissions and

from this, gain insight into how significant these parameters are for the radiative effects in the models. The goal is to better understand the processes controlling sensitivities in common setups of SOA parameterizations in ESMs currently. We do not conclude on whether these processes are treated correctly, but rather if it is important that they are.



## 2 Method

### 2.1 Experimental Setup

A set of sensitivity experiments were designed to investigate how changes to BVOC and SOA representations in the models affect clouds and radiation balance. Care was taken to design experiments that could be run with all three models. The 5 sensitivity experiments are:

- **Yield higher** - The EL/L/SVOC yields for the BVOC oxidation reactions (Table 2) are increased by 50 %.

- **Yield lower** - The EL/L/SVOC yields for the BVOC oxidation reactions are decreased by 50 %.

- **No ELVOCs** - The formation of ELVOCs is removed from the models. The total BVOC oxidation yields are kept constant but all BVOC oxidation reactions produce L/SVOCs.

- **No isoprene** - the isoprene emissions in the models are turned off.

- **No MTs** - the MT emissions in the model are turned off.

For comparison purposes, a control simulation (CTRL) was run with the models. The *Yield higher* and *Yield lower* simulations
directly increase or decrease the produced SOA mass, while the changes are more complex for the other 3 experiments. The *No ELVOCs* scenario strongly decreases the NPF in the BL, and increases the mass of L/SVOCs which can only condense onto existing particles. With the *No isoprene* and *No MTs* cases, the importance of the two classes of BVOCs for modelled SOA mass and particle size distributions is investigated. Whereas isoprene is, on a global scale, emitted in larger amounts than MTs, its' oxidation reactions have smaller yields for SOA precursors than MTs. In addition, the modelled isoprene oxidation
produces very little (EC-Earth, ECHAM) or no (NorESM) ELVOCs. Therefore, with these two experiments both the amount of modelled SOA and the fraction of oxidation products participating in NPF are changed.

In order to have similar meteorological conditions in the three models, all simulations were nudged (Kooperman et al., 2012) to ERA-Interim (Dee et al., 2011) data for the years 2000 - 2005. Although this method may not capture all changes in the cloud radiative effect (CRE) since dynamical feedbacks are limited by the constrained meteorology (Lohmann and Hoose, 2009; Lin
et al., 2016), previous studies with CAM5.3-Oslo found the effective radiative from aerosol-cloud interactions (ERFaci) and ERFaci changes carried out with nudged configurations to be in the uncertainty range of that carried out with a free running version of the model (Kirkevåg et al., 2018; Karset et al., 2018). The first year of the simulations has been discarded as a spin-up and the last 5 years have been used for the analysis.

The radiative effects from aerosols and clouds in this study are calculated using the methods described by Ghan (2013).
The direct radiative effect (DRE) is calculated by taking the difference between the top of the atmosphere radiative flux and the radiative flux excluding scattering and absorption by aerosols ($F_{clean}$). The CRE is similarly calculated as the difference between $F_{clean}$ and the radiation flux without the scattering and absorption by the clouds or aerosols ($F_{clean,clear}$).





## 2.2 Model similarities and dissimilarities

A detailed description of each of the models will follow after this section. However, here we would like to highlight some

of the key similarities and dissimilarities between the models (summarized in Table 1). The ESM model components are different between the three ESMs. The Norwegian Earth System Model (NorESM) and ECHAM have an atmospheric model which contains an aerosol module while EC-Earth consist of a chemistry transport model coupled to an atmospheric general circulation model. Therefore, EC-Earth has a more advanced treatment of gas-phase chemistry, including interactive oxidant fields, while the other two models have prescribed oxidant fields. The aerosol modules also differ between the models. NorESM

has OsloAero (Kirkevåg et al., 2018) while EC-Earth and ECHAM both use M7 (Vignati et al., 2004), but different versions. As described in the Introduction, the treatment of the oxidation products is similar between the models where all three models have one ELVOCs and one L/SVOC tracer. However, the BVOC oxidation differs between the models in terms of yield, number of oxidation reactions from BVOC to ELVOCs/L/SVOCs and which reactions produce which oxidation products, as can be seen in Table 2. Moreover, all three ESMs use the Model of Emissions of Gases and Aerosols from Nature (MEGAN) (Guenther

et al., 2012) but forced with different underlying vegetation and meteorology, thus still generating different emissions in the different models.

## 2.3 NorESM

NorESM (Bentsen et al., 2013; Kirkevåg et al., 2013; Iversen et al., 2013) is an ESM based on the Community Earth System Model (CESM) (Neale et al., 2012). The aerosol scheme in the atmospheric community model (CAM) version 5.3 has been

replaced by the aerosol scheme OsloAero (Kirkevåg et al., 2018). In this investigation, CAM5.3-Oslo (CAM with OsloAero) is coupled to the community land model (CLM) version 4.5 (Oleson et al., 2013) run with satellite phenology (SP) vegetation. NorESM is run with prescribed sea surface temperature (SST) and sea ice concentrations at 1.9 x 2.5° resolution. The horizontal winds and surface pressure are nudged to ERA-interim data with a relaxation time of 6 hours.

OsloAero is described as a production tagged aerosol model which consists of 12 lognormally shaped background modes. The

shape and size of these modes can be modified by coagulation and condensation. The modes are made up of background tracers which determine the number concentration and process tracers which change the shape of aerosol size distribution. The mass of the tracers is tracked and the size distributions and optical properties are calculated using a look-up table approach (Kirkevåg et al., 2018).

NPF was recently added to OsloAero (Makkonen et al., 2014) and is now included as two background tracers, one for sulphate

($SO_4$) and one for SOA, forming one mode (Kirkevåg et al., 2018). Two types of NPF are included in OsloAero; (1) Binary homogeneous sulphuric acid-water nucleation according to Vehkamäki et al. (2002) and, (2) an activation type nucleation, in the BL, with a nucleation rate calculated from Eq. 18 in Paasonen et al. (2010). This nucleation rate is calculated from the concentrations of $H_2SO_4$ and ELVOCs available for nucleation. The nucleation rates are calculated for particles with a diameter of 2 nm but the diameter of the nucleation tracers in OsloAero is 23.6 nm. The survival of these newly formed

particles from nucleation to 23.6 nm diameter is parameterized dependent on coagulation sink and condensation growth rate in




accordance with Lehtinen et al. (2007). In this study, contrary to Kirkevåg et al. (2018), we include all pre-existing particles in the calculation of coagulation sink. This modification was introduced into the model in order to have a more realistic survival rate of the particles between 2 and 23.5 nm.

The BVOC emissions used in the simulations are calculated interactively by MEGAN version 2.1 (Guenther et al., 2012) which is included in CLM. MEGAN thus uses the vegetation from CLM. The BVOC emissions depend on factors such as temperature, radiation, leaf area index and soil moisture. The model is run with half an hour time step and the coupling between CLM and CAM-Oslo is done at every time step providing an interactive diurnal variation in the emissions. The BVOC emissions include isoprene and 7 compounds which are lumped together as MTs in CAM-Oslo.

In CAM-Oslo, the emitted BVOCs are transformed into SOA through chemical reactions with ozone, hydroxide (OH) and 160 nitrate ($NO_3$). When MTs reacts with $O_3$, ELVOCs are formed, while the other five reactions yield only L/SVOCs. The reactions and their yields are given in Table 2. 50 % of the formed ELVOCs is available for nucleation and the rest of the ELVOCs and the L/SVOCs condense onto pre-existing aerosol particles (Makkonen et al., 2014). The molar mass of both the ELVOCs and L/SVOCs are 168 g mol$^{-1}$. The oxidants are prescribed monthly fields originating from a run with the full chemistry model CAM-chem (Lamarque et al., 2012).

CAM5.3-Oslo uses the cloud bulk microphysics scheme MG1.5 (Morrison and Gettelman, 2008; Gettelman and Morrison, 2015) with aerosol activation by Abdul-Razzak and Ghan (2000) for the stratiform clouds. Mass and number of cloud water and ice are treated prognostically while the precipitation is diagnostic. The model also includes a shallow convection scheme (Park and Bretherton, 2009) and a deep convection scheme (Zhang and McFarlane, 1995).

## 2.4 EC-Earth

The Earth system model EC-Earth (Hazeleger et al., 2012; van Noije et al., 2014) includes an atmospheric general circulation model (GCM) based on cycle 36r4 of the Integrated Forecasting System (IFS) of the European Centre for Medium-Range Weather Forecasts (ECMWF). This is coupled to the atmospheric chemistry and transport model TM5 (Tracer Model 5, van Noije et al., 2014; Williams et al., 2017). For the present study, the EC-Earth release v3.2.3 in atmosphere-only mode (i.e., IFS + TM5) was used. TM5 treats the emission, transport, microphysical and chemical conversions, and deposition of atmospheric 175 gases and aerosols. The latter are described with the size-resolved modal microphysics scheme M7 (Vignati et al., 2004). It uses seven log-normal size distributions (modes) of which four are soluble (nucleation, Aitken, accumulation, coarse) and three insoluble (Aitken, accumulation, coarse). The nucleation, Aitken, accumulation, and coarse mode represent particles with dry diameters smaller than 10 nm, 10-100 nm, 100 nm - 1 $\mu$m, and larger than 1 $\mu$m, respectively. The considered aerosol species are sulphate, black carbon, organic matter (primary and secondary), mineral dust, and sea salt. M7 tracks the number concentration 180 in each mode and the mass of each species in each mode. The mode shape is constant whereas median diameter, number of particles of each mode and their chemical composition can evolve freely. After growth by condensation and coagulation, the largest particles of each mode are shifted to the next larger mode. Particles in the insoluble modes that reach sufficient soluble coating are shifted to the respective soluble modes.

In addition, TM5 simulates the total particulate mass of nitrate, ammonium, and methane sulphuric acid. When calculating





optical properties these components as well as the associated water uptake are assumed to be in the soluble accumulation mode. The gas-phase chemistry is described by a modified version of the Carbon Bond 05 (CB05) mechanism (Williams et al., 2017) using the photolysis scheme from Williams et al. (2012). SOA is produced from MTs and isoprene with the yields as presented in Table 2. Assumed molar masses for the two SOA species, ELVOCs and L/SVOCs, are 248 g mol$^{-1}$ and 232 g mol$^{-1}$, respectively. Produced ELVOCs condense to soluble nucleation, Aitken, accumulation, and coarse mode as well as

to insoluble Aitken mode according to the respective condensation sink (depending on surface area). L/SVOCs condense to soluble Aitken, accumulation, and coarse mode as well as insoluble Aitken mode according to the actual mode mass.

NPF is treated following Vehkamäki et al. (2002). In addition, NPF from $H_2SO_4$ and ELVOCs is calculated using the semi-empirical method by Riccobono et al. (2014) (see Table 1). The size of freshly nucleated sulphuric-acid-ELVOC-clusters is assumed to be 2 nm. The early growth to 5 nm diameter is calculated following Kerminen and Kulmala (2002). The resulting

number of these 5 nm-particles are finally added to the nucleation mode.

The emission of the SOA precursors isoprene and MTs are calculated using MEGAN-MACC (Sindelarova et al., 2014) for the year 2000 and depend on the underlying vegetation information. The BVOC emissions are prescribed monthly fields and with an applied diurnal variation. There is also small fraction of MTs and isoprene emitted from biomass burning, which will participate in production of SOA.

For the present study, IFS is applied at a spectral truncation of T255 (corresponding to 0.7°) grid with 91 vertical levels. Emissions for TM5 are applied on a 0.5°×0.5° grid whereas following processes and transport are calculated on 3°×2°. The horizontal winds (via divergence and vorticity) and surface pressure were nudged against ERA interim with a relaxation time of 8.25 hours. The cloud droplet number concentration of stratiform clouds is calculated using Abdul-Razzak and Ghan (2000), and both determines the effective radius of the cloud droplets and influences the lifetime of the clouds via its effect on the

autoconversion of cloud liquid water to rain.

## 2.5 ECHAM

ECHAM5-HAM (Stier et al., 2005) is an aerosol-climate model originally developed at the Max Planck Institute for Meteorology, Hamburg. The Hamburg Aerosol Module (HAM) also employs the M7 aerosol microphysics module. The ECHAM-HAM simulations were performed in T63 spectral resolution with 31 hybrid-sigma vertical levels. The spectral atmospheric variables

are nudged with standard ECHAM relaxation timescales: 6 h for vorticity, 24 h for pressure and temperature, and 48 h for divergence (Lohmann and Hoose, 2009).

We apply a modified version of ECHAM5.5-HAM2 (Zhang et al., 2012a), which uses an improved numerical scheme (Kokkola et al., 2009) to compute the formation of sulphuric acid by oxidation of $SO_2$, and its removal by nucleation and condensation on pre-existing particles. We consider SOA formation from the biogenic precursors isoprene and MTs. The SOA forma-

tion mechanism (Jokinen et al., 2015) includes both kinetic condensation to Fuchs-corrected surface area (condensation sink) and partitioning according to pre-existing organic mass. The model considers three BVOC tracers: isoprene, endocyclic and other MTs. The BVOC emissions in ECHAM simulations were pre-computed monthly averages (Jokinen et al., 2015). The MEGAN2.1 (Guenther et al., 2012) was driven with input drivers described in Sindelarova et al. (2014), combining MERRA



meteorological fields and MACC landcover data. However, the BVOC emission inventories did not separate endocyclic and
other MTs, hence their respective emissions were considered equal fractions. The reaction rates of SOA precursors with $O_3$,
OH and $NO_3$ are described in Jokinen et al. (2015).

The ELVOC yields are based on extensive laboratory experiments (Jokinen et al., 2015), while the total (ELVOC+L/SVOC)
yield is set to 15% for MTs and 5% for isoprene. ELVOCs provide early growth for nucleation mode particles, as they are
distributed to the particle phase according to condensation sink. The low- and semi-volatile products are distributed to particle
phase according to particle-phase organic mass, as in Jokinen et al. (2015). Hence, after oxidation, no SOA products remain in
the gas-phase, but immediate condensation to aerosol-phase is assumed. Simulations include organic vapours in the nucleation
process according to Eq. 18 in Paasonen et al. (2010). The growth from nucleation to 3 nm is calculated according to Kerminen
and Kulmala (2002) assuming growth by ELVOCs and sulfuric acid.

## 3   Results and discussions

We will start the results section by investigating the inter-model differences in the CTRL simulation among the three models. It
is necessary to be aware of the differences between models before investigating at the changes that the sensitivity simulations
induce.

### 3.1   CTRL

#### 3.1.1   BVOC emissions and concentrations

The three ESMs all use the same emission model (MEGAN) but the emissions of MTs and isoprene still vary between the
models because of choices in land cover data and meteorology. For isoprene, NorESM has the lowest emissions rates of about
435 Tg yr$^{-1}$ while EC-Earth and ECHAM are somewhat higher with 572 and 526 Tg yr$^{-1}$ , see Fig. 1g. The spatial distribution
in the emissions also varies between the models. In NorESM, the isoprene emissions are highest in the Amazon region with
somewhat smaller sources in Africa and the tropical islands of Indonesia (Fig. 1a). EC-Earth has the highest emission rates out
of all models but with the Amazonian maximum located further south than in NorESM (Fig. 1c). ECHAM has similar emission
patterns to EC-Earth but with somewhat weaker emissions, see Fig. 1e.

For MTs, NorESM has the highest global emissions (118 Tg yr$^{-1}$) followed by EC-Earth (96 Tg yr$^{-1}$) and then ECHAM (77
Tg yr$^{-1}$) as can be seen in Fig. 1h. The largest differences in the emissions are in the tropics, in particular in the Amazonian
region, where NorESM has up to twice as high annual emissions. The cause of the difference in emissions is related to the
implementation of MEGAN used in the models. In NorESM, MEGAN is interactive and uses the vegetation from CLM as well
as atmospheric conditions and radiation in the calculation of the BVOC emissions. EC-Earth and ECHAM on the other hand
use prescribed BVOC emissions from MEGAN-MACC with a yearly as well as diurnal variation included (Sindelarova et al.,
2014). The emissions from these two models are not the same because different meteorology was used in the generation of the





emissions fields.

The column burdens of the BVOCs also differ between the models (not shown). The global mean column burden of isoprene is approximately three times higher in EC-Earth ($1.0 \, \mathrm{kg \, m^{-2}}$) than in the other two models. This is in part because of significantly higher column burdens over the strong emission regions in South America and Africa, which are due to the interactive oxidant fields in EC-Earth. When interactive oxidant fields are used the oxidants can be depleted and as a result the lifetime of BVOCs

is increased. This does not occur in the other two models that have prescribed oxidation fields. The MT column burdens are more similar between the models.

### 3.1.2 SOA formation and aerosol size distributions

There is a large range in the amount of SOA formed in the different models. In spite of having the lowest BVOC emissions (due to lower isoprene emissions), NorESM has the largest average annual production ($85 \, \mathrm{Tg \, yr^{-1}}$) while ECHAM and EC-

Earth have very similar and somewhat lower SOA production ($52 \, \mathrm{Tg \, yr^{-1}}$), see Fig. 2a. The higher emissions in NorESM is likely a result of the higher MT emissions (which have the highest yields), in combination with higher yields for isoprene than EC-Earth. Also the assumed molar mass of the BVOC oxidation products will affect how much SOA mass is formed.

In this paper we have averaged the size distribution and number concentration data globally over the model levels with pressures higher than 850 hPa, i.e the bottom part of the atmosphere. This choice was made since this part of the atmosphere contains most

of the aerosol mass which is relevant for both the direct and indirect aerosol effects. Moreover, clouds in the ESMs use aerosol activation at the bottom of the clouds and thus, the aerosols at these levels are most important also for the indirect aerosol effects. The aerosol size distributions (for all particles) show large differences between the models even though ECHAM and EC-Earth both use the modal aerosol model M7, see Fig. 3. The most noticeable difference between the number size distribution of the models is that NorESM, which uses the aerosol model OsloAero, has no explicit nucleation mode. In NorESM, particles

from NPF are added directly into $SO_4$/SOA nucleation mode which is in the Aitken-accumulation size range after growth through condensation. ECHAM's size distribution is dominated by a large nucleation mode which contains almost two orders of magnitude more particles than the nucleation mode in EC-Earth. Moreover, EC-Earth also has fewer particles than the other models in the largest particle sizes (diameters > 250 nm). Of the three models, ECHAM has the most particles at large sizes (diameters >300 nm) as well as highest surface and volume of particles (Fig. 3 b and c). Total aerosol number concentrations

(Fig. 4) reveal that EC-Earth has the lowest aerosol number concentrations out of all models and ECHAM has the highest. This is still the case when comparing the number concentrations without the nucleation mode. Moreover, ECHAM has substantially higher aerosol number concentrations over the remote oceans (Fig. S1) compared to the other two models.

### 3.1.3 AOD and direct aerosol effects

The global average aerosol optical depth (AOD) is highest in NorESM, 0.19, (Fig. 2b) due to significantly higher AOD values

over desert regions, in particular the Sahara (Fig. S1). ECHAM has the second highest global AOD values (0.16) and has somewhat higher AOD values over the ocean than the other two models. The direct aerosol effects (Fig. 2c) in the models resemble the results from the AOD. EC-Earth has lower globally averaged direct radiative effect ($\mathrm{DRE}_{Ghan}$) than the other





two models. This is a result of the low aerosol number concentrations, in particular at larger, radiation relevant sizes. This can also be seen in the AOD from EC-Earth. ECHAM has slightly stronger global average $DRE_{Ghan}$ than NorESM even though

NorESM has a higher average AOD. The reason for this is that many of the regions with large AOD in NorESM have very bright surfaces (e.g. deserts) and therefore result in a lower $DRE_{Ghan}$ (Fig. S1).

### 3.1.4  Cloud properties and indirect aerosol effect

The cloud properties in CTRL simulation are quite different in the models. EC-Earth has the lowest cloud droplet number concentrations (Fig S2) which is related to the low number concentrations of aerosol particles in this model. ECHAM on the other

hand has the highest number of cloud condensation nuclei (CCN) and also the highest cloud droplet number concentrations (CDNC). NorESM has larger droplet sizes than the other two models and the droplet size patterns are very different in the different models. Nevertheless, the total grid box cloud water path (CWP) is fairly similar between the three models, but slightly higher in EC-Earth (Fig. 2d). The total cloud fractions (CF) in the models are also fairly similar with global average values between 0.61 and 0.67, see Fig. 2e. The cloud radiative effect ($CRE_{Ghan}$), is stronger in NorESM (-31 W m$^{-2}$) compared

to EC-Earth (-23 W m$^{-2}$) and ECHAM (-26 W m$^{-2}$), see Fig. 2f. Note that these are development version of NorESM and EC-Earth which has not been tuned. The patterns of the CF and $CRE_{Ghan}$ can be seen in Fig. S3.

### 3.2  Yield higher and Yield lower

The results from the sensitivity simulations will now be presented and discussed in three different sections. The sensitivity experiments are grouped according to the similarity in the results. In the first section, the *Yield higher* and *Yield lower* experiments

are discussed.

### 3.2.1  Direct aerosol effects

First, the results regarding the changes in aerosol scattering and how these affect climate forcing are presented. In the *Yield higher* simulation the $DRE_{Ghan}$ becomes stronger, i.e. more negative, and the opposite is true for the *Yield lower* simulation, for all three models (Fig. 5). These changes reflect the changes in SOA formation (Fig. 6) as more SOA leads to a stronger

$DRE_{Ghan}$. Since NorESM has the largest SOA production it also experiences the largest SOA production change in these simulations, approximately $\pm$ 38 Tg per year. The changes in the other two models are in the order of 25 Tg per year. Interestingly, an increase / decrease in the SOA precursor yields by 50 % results in an increase / decrease in SOA production by 50 % only in EC-Earth. In NorESM, the SOA production change is somewhat less than 50 % in both simulations. The explanation for this is that SOA is also produced from dimethyl sulphide (DMS) emissions from the ocean in NorESM (Kirkevåg et al., 2018),

and these yields are not changed in the sensitivity simulations. For ECHAM, the effect is somewhat larger in the *Yield higher* simulation (+52 %) and smaller in the *Yield lower* simulation (-45 %).

The degree to which a SOA increase leads to an strengthening in the $DRE_{Ghan}$ varies between the models. NorESM has the largest absolute increase / decrease in SOA formation in these two simulations but it is EC-Earth that experiences the largest





change in the $\text{DRE}_{Ghan}$ with changes of $\pm$ 0.15 $\text{Wm}^{-2}$. For reference, this number is roughly half of the radiative forcing

due to aerosol–radiation interactions (RFari) best estimate in the Fifth Assessment Report by the IPCC (2013). ECHAM has

the smallest changes in $\text{DRE}_{Ghan}$ with values of approximately $\pm 0.03$ even though this model has similar changes in SOA

production changes to EC-Earth. The cause of the different responses in the different models is, at least partly, related to where

in the aerosol size distribution the additional / removed SOA is located. For all three models in the *Yield higher* simulation,

the globally averaged particle number concentrations increase at sizes relevant for scattering of solar radiation ($\text{N}_{d>100}$, num-

ber concentration of particles above 100 nm) (see Fig. 4). In ECHAM however, this increase is quite small, see Fig. 4f. The

changes in particle number concentration in NorESM are quite large but are mainly located close to the BVOC sources (not

show). EC-Earth instead experiences these changes in the particle number concentration further downwind of the sources. This

results in a more widespread change in $\text{DRE}_{Ghan}$ in EC-Earth compared to the other two models (see Fig. S4) and thus a

significantly higher global average $\text{DRE}_{Ghan}$. Similar but opposite changes are seen in the *Yield lower* simulation.

**3.2.2  Indirect aerosol effects**

The response of the indirect aerosol effects in the *Yield higher* and *Yield lower* sensitivity test differ more than the direct

effects. The $\text{CRE}_{Ghan}$ in NorESM is strengthened (i.e. more negative) with increasing SOA production and vice versa, see Fig.

7. The changes in $\text{CRE}_{Ghan}$ are -0.27 W $\text{m}^{-2}$ (*Yield higher*) and 0.35 W $\text{m}^{-2}$ (*Yield lower*) indicating that these sensitivity

simulation induce changes in the forcing of relevant magnitude. The globally averaged changes in EC-Earth have the same sign

as those for NorESM but are lower (-0.11 and +0.076 W $\text{m}^{-2}$), and for ECHAM the changes are very small and not statistically

significant. Also for the indirect effects, changes in the size distributions can be used to explain the changes in $\text{CRE}_{Ghan}$. While

hygroscopisity might play a role, the effect is small in the activation scheme shared by the models (Abdul-Razzak and Ghan,

2000).

For NorESM, the higher (lower) SOA production in the *Yield higher* (*Yield lower*) simulation results in a shift in the size

distribution to larger (smaller) sizes, see Fig. 4. For the *Yield higher* simulation, this results in higher CCN concentrations,

higher CDNC, smaller cloud droplet effective radius ($\text{r}_e$) and smaller CWP (Fig S4 - S7). The opposite change in these variables

is seen in the *Yield lower* simulation. The main relative changes in cloud variables in NorESM are located over and downwind

of the large BVOC emission sources in the Tropics. Increased number of CCN generally means higher CDNC, lower $\text{r}_e$ and

higher CWP in all three models. For some regions, the CF decreases as CCN increases. The results regarding the changes in

cloud parameters are shown in Fig. S5 - S8 for the *Yield higher* and *Yield lower* simulations. Since the cloud response to the

CCN changes are similar in all the models and simulations, we will mainly discuss CCN and $\text{CRE}_{Ghan}$ changes for the other

simulations.

The essentially non-existing effects on the $\text{CRE}_{Ghan}$ in ECHAM can also be explained using the size distribution. ECHAM

experiences the smallest changes in particle number concentrations for particles with diameters > 100 nm, see Fig. 4f. However,

for the smallest size ranges ($\text{N}_{1-60}$) ECHAM has the largest changes in the concentration of particles. Not surprisingly, the

changes in number of these small particles do not affect the cloud formation in ECHAM, probably because they are too small

to act as CCN when there is an abundance of particles at larger sizes (accumulation mode). Another interesting feature of





ECHAM is that the changes in the size distribution are not mirrored in *Yield higher* and *Yield lower* simulation, which suggest non-linear dynamics caused by competition between NPF and condensation sink.

The rather small global changes in $CRE_{Ghan}$ for EC-Earth are somewhat surprising since this model had the strongest response for the $DRE_{Ghan}$. However, investigating the maps of the changes in the $CRE_{Ghan}$ for EC-Earth in Fig. 8 c-d, one can see that the low global responses are caused by a pattern of opposite changes with magnitudes up to 4 Wm$^{-2}$. In the *Yield higher* simulation, there is a strengthening of the $CRE_{Ghan}$ close to large BVOC emission regions in the Tropics, while over the remote oceans there is a weakening instead. The mirrored response is seen in the *Yield lower* simulation. Since the SOA production

increases (decreases) globally in the *Yield higher* (*Yield lower*) simulation, the opposing patterns of $CRE_{Ghan}$ are not directly related to changes of SOA production. Instead, the changes are related to different effects on the size distribution close to and far away from the BVOC sources as can be seen in Fig. 9 (the areas are shown in Fig S21). For the *Yield higher* simulation, close to the sources, the increase in SOA production results in more accumulation mode particles ($N_{100-500}$), which leads to higher CCN concentrations and a stronger $CRE_{Ghan}$. Over the remote regions there is also an increase in accumulation mode

particles, but this is accompanied by a larger decrease in particle concentrations in the Aitken mode ($N_{20}$ - $N_{60}$). Since the aerosol concentrations are low in EC-Earth, in particular in these remote regions, the particles in the Aitken mode can also be activated as CCN because reduced competition effects gives higher maximum super saturation during cloud droplet activation. As a result, the CCN concentrations in these remote regions decrease when the SOA formation increases. This leads to a weakening of the $CRE_{Ghan}$ (positive values). The changes in the *Yield lower* mirror those in the *Yield higher* simulation.

## 3.3   No ELVOCs

In this second section of results from the sensitivity simulations the results from the *No ELVOCs* simulation are presented. This simulation is different from the other simulations since only the type of SOA precursors is changed and not the amount of precursors.

### 3.3.1   Direct aerosol effects

In terms of the direct effects, the global changes are small in all three models. For NorESM there is a small but statistically significant strengthening of the $DRE_{Ghan}$, but the other two models do not display significant changes (see Fig. 5). The change in NorESM can be explained by changes in aerosol number concentrations over and downwind of the Amazon. Since there is no ELVOCs contributing to nucleation in this simulation, the NPF is reduced, and with this the number concentration of smaller particles. This decrease is particularly strong over the Amazon since the MT emissions are very high here (see Fig

1b) and ELVOCs can only be produced from MTs in NorESM. The strong decrease in small particles and increased vapours available for condensation (L/SVOCs) in this region means that more particles can grow to sizes where they act as efficient scatterers of solar radiation. This effect over the Amazon in NorESM is big enough to affect the global $DRE_{Ghan}$.

NorESM produces a large number of particles close to the BVOC emissions sources and, since the model does not contain a nucleation mode these particles are introduced into the Aitken mode. The nucleated particles thus reach larger sizes closer to

the sources than in other two models where the particles are introduced into a nucleation mode and shifted to the Aitken mode





at a later time step, while they are transported. This could be part of the explanation of why the BVOC effects in NorESM, in general, are located closer to the sources than in EC-Earth. Moreover, in comparison to NorESM, EC-Earth has lower oxidant concentrations close to the large BVOC sources (not shown), which limits the SOA production in these regions and increases the amount of BVOC transported away from the sources. Hence, the overall effect is more widespread than in NorESM.

### 3.3.2 Indirect aerosol effects

For the indirect aerosol effects, EC-Earth is the only model that has significant changes for this simulation. The $CRE_{Ghan}$ is weakened (less negative) by 0.44 $Wm^{-2}$ as can be seen in Fig. 7. This strong change in the $CRE_{Ghan}$ is caused by a more or less strong worldwide decrease in the aerosol number concentration at almost all sizes (except particles above 500 nm). This results in a reduction of CCN which leads to a weakened $CRE_{Ghan}$. This strong decrease in CCN in EC-Earth occurs since the nucleation rates involving ELVOCs are calculated from a product of the $H_2SO_4$ and ELVOC concentrations (see Table 1). Thus, the removal of ELVOCs in this simulation removes all the BL NPF in EC-Earth. The other two models instead calculate the nucleation rates as the sum of $H_2SO_4$ and ELVOC concentrations and thus retains BL NPF from $H_2SO_4$. This results in quite different spatial patterns of the reduction in total aerosol number concentrations in the different models. For NorESM and ECHAM, the reductions occur close to the BVOC sources. For EC-Earth on the other hand, the reductions are largest over regions that have large anthropogenic $SO_2$ emissions such as Europe, North America and Australia (not shown). This widespread reduction in CCN in combination with EC-Earth having low aerosol concentrations (which makes the clouds more sensitive to aerosol perturbations (Spracklen and Rap, 2013)) results in a significant weakening of the $CRE_{Ghan}$ in this simulation.

ECHAM has no significant change in the $CRE_{Ghan}$ when the ELVOCs are removed. As for the first two experiments, the changes in the particle concentrations in the accumulation mode are small (Fig. 4). ECHAM, unlike the other models, experience an increase in nucleation mode particles in this simulation. This is somewhat unexpected since removal of ELVOCs is expected to result in a decrease in NPF. This simulation however shows that the nucleation rate parameterization in ECHAM is not very sensitive to ELVOC concentrations. Nevertheless, the growth of the newly formed particles is highly dependent on the ELVOC concentrations and since the particles do not grow to larger sizes, more particles remain in the nucleation mode. This results in increasing the concentration in the nucleation mode and decreasing number concentrations at larger sizes.

### 3.4 No isoprene and No MTs

In this last result section the *No isoprene* and *No MTs* simulations will be shown and discussed.

### 3.4.1 Direct aerosol effects

The $DRE_{Ghan}$ is reduced (less negative) in all models in both these simulations since the SOA formation goes down when the BVOC emissions are reduced. The strongest effects on the $DRE_{Ghan}$ is seen in EC-Earth with approximately 0.15 $W\,m^{-2}$ changes for the *No isoprene* and *No MTs* simulations (Fig. 5). The reductions in $DRE_{Ghan}$ for NorESM are about twice as large





as those for ECHAM for the *No MTs* simulation. Moreover, both NorESM and ECHAM have almost an order of magnitude smaller decreases than in EC-Earth. In NorESM and ECHAM the changes in $DRE_{Ghan}$ are located fairly close to the sources while in EC-Earth they have a larger geographical spread (Fig. S15). This is the main cause for the large changes in the global

$DRE_{Ghan}$ in EC-Earth. The decrease in $DRE_{Ghan}$ is explained by a reduction in the concentration of particles relevant for scattering (diameters above 100 nm).

The difference in SOA production between these cases reflects the proportion of SOA originating from isoprene and MTs respectively. NorESM and ECHAM have the largest reductions in the *No isoprene* case, indicating that isoprene is the dominant SOA precursor, while EC-Earth has the largest reduction in the *No MTs* simulation, indicating MTs are the dominant precursors.

The difference in dominating precursors in the different models is mainly a result of different yields. The EC-Earth isoprene yields are 1 % (ELVOCs + L/SVOCs), 15 times lower than for MTs, while in NorESM and ECHAM, the isoprene yield is 5 %, 3 times smaller than for MTs.

In the *No isoprene* simulation we also see an interesting feature connected to the interactive gas-phase chemistry in EC-Earth. Over large emission regions in the Tropics, the column burden of MTs decreases when we remove the isoprene emissions, see

Fig 10. The decrease in the MT column burden is caused by a greater availability of OH when there is no isoprene present. $O_3$ is reduced, but is less important for the loss rate. This occurs only in EC-Earth due to the interactive chemistry in TM5. This does not increase the amount of SOA formed from MTs but it affects where this SOA is formed, causing formation of SOA to occur closer to the sources. Additionally it favours the L/SVOCs over ELVOCs (see Table 2) because oxidation with OH will dominate more over reactions with $O_3$, and MT oxidation with $O_3$ have a higher ELVOC yield (5%) than with OH (1%).

### 3.4.2   Indirect aerosol effects


The *No isoprene* simulation displays the largest and also the most divergent results out of all simulations for the indirect aerosol effects. NorESM has a weakened $CRE_{Ghan}$ by 0.53 W m$^{-2}$ while EC-Earth has a strengthened $CRE_{Ghan}$ by -0.82 W m$^{-2}$. These numbers show that there is a substantial impact on the $CRE_{Ghan}$ from the isoprene emissions in NorESM and EC-Earth. The magnitude of these numbers are in the range of and larger than the best estimate of the IPCC (2013) ERFaci relative to

1750 of -0.55 W m$^{-2}$. ECHAM on the other hand experience a non-significant change in $CRE_{Ghan}$ in, see Fig. 7. Interestingly, all three models show a somewhat similar change in the size distribution as can be seen in Fig. 4. Isoprene mainly produces L/SVOCs in the models and the removal of isoprene therefore leads to a shift in the particle size distribution towards smaller particles. For ECHAM, the aerosol concentration changes at CCN relevant sizes are very small and the clouds are virtually unaffected by this change. NorESM on the other hand, experiences a quite large decrease in accumulation mode particles

which results in a decrease in CCN and weakening of the $CRE_{Ghan}$. EC-Earth also experience a decrease in particles above 100 nm. Moreover, the Aitken mode in EC-Earth has a large absolute increase. This is due to more NPF when the condensation and coagulation sink decreases (more on this in the next section). Since the aerosol number concentrations in EC-Earth are so low, even aerosol particles in the Aitken mode can be activated as CCN and the increase in Aitken mode particles leads to increased CCN concentrations. This results in a strengthening of the $CRE_{Ghan}$, in particular over the oceans (Fig S20). Thus,

similar changes in the size distribution lead to vastly different responses in the three models depending on aerosol number





concentrations and different size distribution dynamics.

For the *No MTs* simulation, EC-Earth and NorESM have a weakened $CRE_{Ghan}$ while ECHAM experiences no significant change in the $CRE_{Ghan}$. The MT oxidation is the main source for ELVOCs in all 3 models (the only source in NorESM). It was therefore expected that turning off the emission of MTs would reduce the amount of small particles in the models. However,

this behaviour is only seen in EC-Earth where the global mean particle number concentration decreases at all sizes globally (see Fig. 4). For NorESM, the number of small particles instead increases while the number of larger particles decreases (due to less condensational growth of the particles), indicating that the loss of L/SVOCs (condensation) from MTs is more important than the loss of ELVOCs (NPF) for the size in this model. In ECHAM the number of nucleation mode particles increases and the number of larger particles decreases, which reduces the sink for small nuclei. Similarly to the *No ELVOCs* simulation

the reduction in ELVOCs leads to limited growth of the nucleation mode particles and therefore, an increase in this mode. However, the changes in ECHAM are again very small and do not affect the clouds. Both NorESM and EC-Earth experience a decrease in CCN and therefore a weakened $CRE_{Ghan}$. The global $CRE_{Ghan}$ response in EC-Earth in this simulation is, as for the simulations with changed yields, a result of compensating opposite patterns of $CRE_{Ghan}$ close to (weakening) and far away (strengthening) from the sources.

**3.5    Further discussion and implications**

The introduction of particles from NPF should in theory and in the models be dependent on the interplay between available vapours for nucleation, condensation losses of these vapours and loss of newly formed particles due to coagulation. Adequately parameterizing these processes is a challenge and the balance between them varies between the models, and also sometimes between regions in the same model. Regional variation is seen EC-Earth where the NPF response varies depending on the

distance from the sources, even if both regions experience the same sign in SOA production change. Taking the *Yields lower* experiment as an example: Close to the sources, the decrease in VOCs lead to a reduction in both larger particles and NPF (Fig. 9). In remote regions on the other hand, the coagulation sink for newly formed particles is reduced because of a reduction in larger particles. This increases the probability of NPF particles surviving to larger sizes. Thus, even though the total aerosol mass is decreased, the mass is partitioned to smaller sizes and the total number concentration is increased. Finally, since EC-

Earth generally has low particle number concentrations in these regions, even these smaller particles are activated to form cloud droplets and produce a negative CRE. A similar effect can be seen in the *No isoprene* case for EC-Earth where we also see a strong negative $CRE_{Ghan}$ associated with an increase in number concentrations (in spite of a decrease in total mass), see Fig. 7 and 6.

The above example for EC-Earth raises a more general point: The relationship between SOA production and CCN and aerosols

relevant for radiation is highly non-linear. If ELVOCs are important for NPF and early growth, then an increase in ELVOCs could lead to more particles formed, but also less condensate to grow the existing particles to climate relevant sizes (CCN, direct radiation effects). On the other hand, if $H_2SO_4$ is driving the NPF, SOA might be more important through changing the coagulation sink for NPF, and more SOA could lead to less NPF, and the effect on CCN will depend on the particles that are left. Thus, NPF does not necessarily lead to higher CCN concentrations.





Another factor of importance to NPF impact on the size distribution is the size at which new particles are added to the aerosol scheme. In EC-Earth and ECHAM, the particle growth and survival to 3 nm (ECHAM) and 5 nm (EC-Earth) are parameterized separately, and then the particles are added to the nucleation mode. The size at which these particles are added make a difference for the transferal of particles to the Aitken mode: in EC-Earth the added particles at 5 nm are already above the number median diameter of the mode and thus some of these will always be transferred to the Aitken mode. In ECHAM on the other hand,

the addition of newly formed particles to the nucleation mode will decrease the number median diameter of the mode and can even decrease the number of particles that are transferred to Aitken mode. If NPF is continuously high, the particles can thus even be inhibited to grow to larger sizes. How much the radius of modes are allowed to change in combination with adding the particles at different sizes could be part of the explanation why EC-Earth and ECHAM show such different changes in aerosol size distributions even though both models use the M7 aerosol module. NorESM also has a separate parameterization for the

growth and survival of the NPF particles up to a radius of 23 nm when the particles are added to the tracer for NPF. This growth occurs in one time step of the model (30 min). Hence, the particles grow very rapidly and reach Aitken mode sizes close to the sources.

There are clearly large differences in the aerosol size distributions and how the changes in these sensitivity experiments affect the size distributions in the models. This is in spite of quite similar simplified treatments of SOA formation in the three models.

Our findings show that the location of the SOA mass in the size distribution is critical for CCN concentrations, which agree with the results in Riipinen et al. (2011). The present study implies that further model development and evaluation is needed in terms of how new particle and SOA formation affect the size distribution. However, there are still large uncertainties on how these models should behave with regards to these processes (Glasius and Goldstein, 2016; Riipinen et al., 2011).

Over the past years, more and more studies have investigated the BVOC climate impact from pre-industrial to present day

(Heald and Geddes, 2016; Scott et al., 2017; Unger, 2014) and also into the future, including possible BVOC climate feedbacks (Sporre et al., 2019; Makkonen et al., 2012; Scott et al., 2018; Paasonen et al., 2013). The results regarding the BVOC impact on climate have a large spread among the different studies. This study indicates that at least parts of these differences could be related to varying sensitivity to BVOC and SOA changes in the models used in the different studies. The decrease in isoprene emissions since 1850 has been estimated to be approximately 15 % (Scott et al., 2017; Unger, 2014) and in this study, the

removal of all isoprene emissions leads to a change in the total aerosol radiative effect by 0.62 W m$^{-2}$ in NorESM and -0.67 W m$^{-2}$ in EC-Earth, a 1.29 W m$^{-2}$ difference. Hence, assuming the changes in radiative effects are not too far from linear, the decrease in isoprene emissions since pre-industrial would introduce an uncertainty in the aerosol forcing of order of magnitude 0.19 W m$^{-2}$ using these models. This sensitivity study reveals that NorESM, EC-Earth and ECHAM would produce very different results if used to investigate the climatic impacts of BVOCs.

**4 Conclusions**

The impact of BVOC emission and SOA formation on particle size distribution, cloud properties and radiative effects have been compared among three ESMs, NorESM, EC-Earth and ECHAM. In five different sensitivity studies, the effect of changed





yields of BVOC oxidation, volatility of the oxidised BVOCs, contribution of precursor gases has been investigated.

We found that both the direct and indirect aerosol effects in the models are substantially affected by changes in SOA precursor

yields and BVOC emissions. The $DRE_{Ghan}$ is strengthened (by up to $0.15\ Wm^{-2}$) by more SOA and vice versa. Even though the changes in $DRE_{Ghan}$ have the expected sign of the response to changes in SOA production in all three models and simulations, the sensitivity of the $DRE_{Ghan}$ to SOA production changes varies between the models. This is connected to how much of the SOA production changes affect the parts of the modelled size distributions where the particles act most efficiently as scatterers of solar radiation. The results from this study show that EC-Earth is the model with most widespread changes of the

accumulation and coarse mode particles, and hence largest sensitivity of $DRE_{Ghan}$. ECHAM is least sensitive here since the SOA changes mostly affect the small particles and are relatively small.

The changes in the $CRE_{Ghan}$ are stronger (up to $-0.82\ Wm^{-2}$) than for the $DRE_{Ghan}$ and more complex. The $CRE_{Ghan}$ changes do not necessarily follow the SOA changes and can be of different sign for different models, and even different regions in the same model. Again, size distribution dynamics are crucial for understanding the sensitivity of the cloud properties

and $CRE_{Ghan}$ in the models. Also for the $CRE_{Ghan}$, ECHAM is the least sensitive model. Overall, the small effects on the size distributions at CCN relevant sizes in this model means that the clouds in ECHAM are virtually unaffected by the sensitivity simulations. The clouds in NorESM are quite strongly affected by the sensitivity simulations, mainly because of shifts in the size distribution. These shifts are mainly a result of changes in condensational growth and thus, the condensation of L/SVOCs is very important for the climate impact of BVOCs in NorESM. EC-Earth is the most sensitive model out of the three models

also for the cloud effects. This results from a combination of a size distribution quite sensitive to NPF in combination with low aerosol number concentrations in EC-Earth, which makes the clouds sensitive to aerosol perturbations. Moreover, the NPF in EC-Earth is more sensitive to the ELVOC concentrations than the other models since the BL nucleation rate is calculated from the product of the $H_2SO_4$ and ELVOC concentrations while the other two models used the sum of the concentrations (see Tab. 1).

We can conclude that the BVOC treatment in the ESMs is of importance and can introduce substantial uncertainties in aerosol climate effects and forcing. There is need for more development and testing of these parameterizations in ESMs, in particular with respect to how the NPF parameterizations affect the size distributions.

*Author contributions.* M.K.S. performed the model simulations with NorESM. M.K.S and S.M.B did the the data analysis and wrote the manuscript. R.S. performed the model simulations with EC-Earth and wrote parts of the paper. R.M. performed the model simulations

with ECHAM and wrote parts of the paper. I.H.H.K. provided support during the setup NorESM. T.v.N., T.B. and D.O. co-developed the EC-Earth version used during the study and provided support for the EC-Earth simulations. M.K.S, S.M.B., R.S., R.M., I.H.K.K. and T.K.B. contributed with discussions regarding the experimental design and data analysis. All contributors have contributed to the discussions regarding the manuscript.





*Competing interests.* The authors declare that they have no conflict of interest.

*Acknowledgements.* The research leading to these results has received funding from the European Union's Seventh Framework Programme (FP7/2007-2013) project BACCHUS under grant agreement no 603445. This work was supported by LATICE, a strategic research area funded by the Faculty of Mathematics and Natural Sciences at the University of Oslo. This work has been financed by the research council of Norway (RCN) through the NOTUR/Norstore project NN9485K Biogenic aerosols and climate feedbacks. I.H.H. Karset has been financed by the research council of Norway through the project EVA and the NOTUR/Norstore projects (Sigma2 account: nn2345k, Norstore ac-
count: NS2345K). TvN and TB acknowledge funding from the European Union's Horizon 2020 research and innovation programme project CRESCENDO under grant agreement No 641816.



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

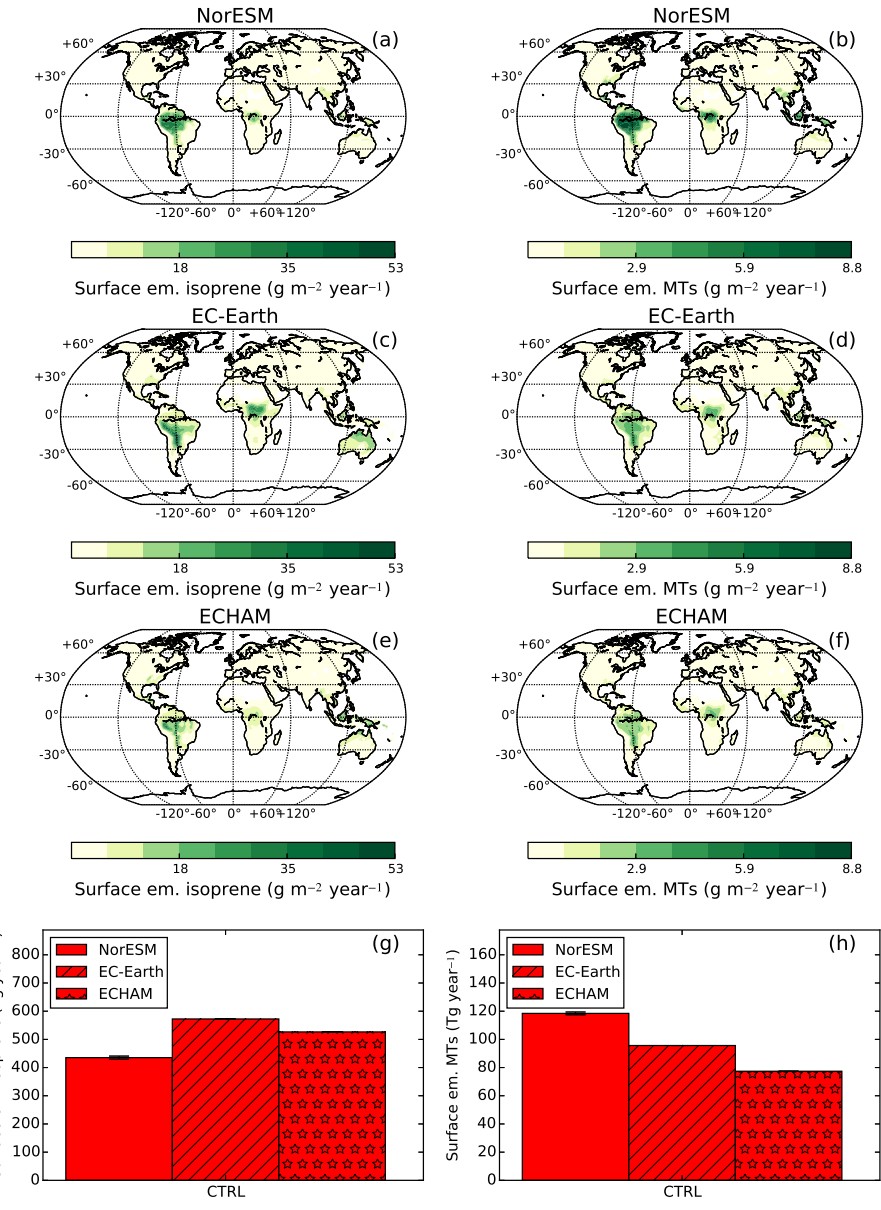

**Figure 1.** Maps of annually averaged surface emissions of isoprene (a,c and e) and monoterpenes (MTs) (b,d and f) for NorESM, EC-Earth and ECHAM. Also shown are the global yearly surface emissions of isoprene (g) and MTs (h). The error bars denote the standard error of mean of the yearly averages.

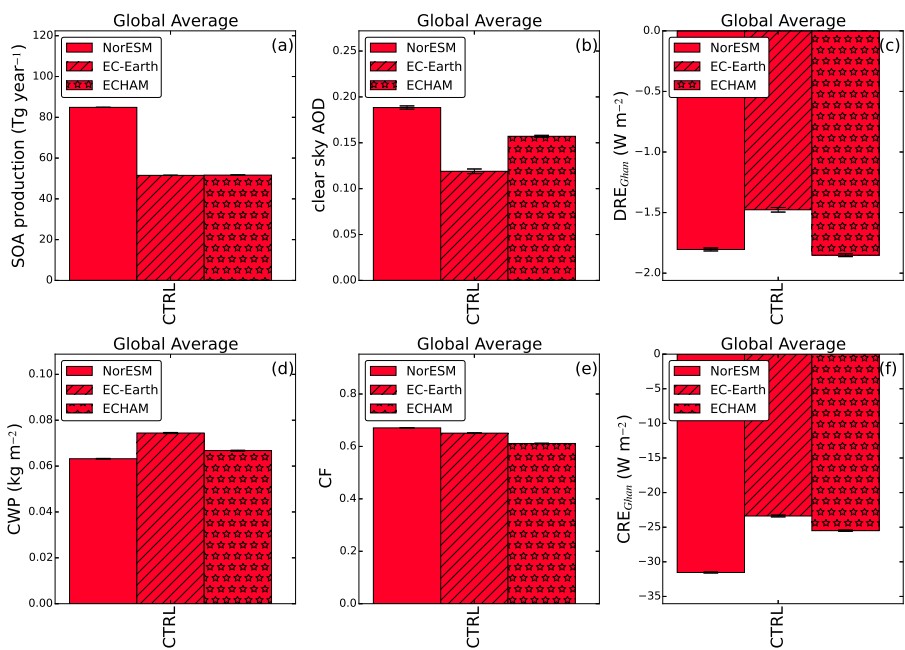

**Figure 2.** Bar plot of the total yearly global SOA production and the yearly averaged aerosol optical depth (AOD), direct radiative effect (DRE$_{Ghan}$), cloud water path (CWP), cloud fraction (CF) and cloud radiative effect (CRE$_{Ghan}$) for all three models for the *CTRL* experiment. The error bars denote the standard error of mean of the yearly averages.



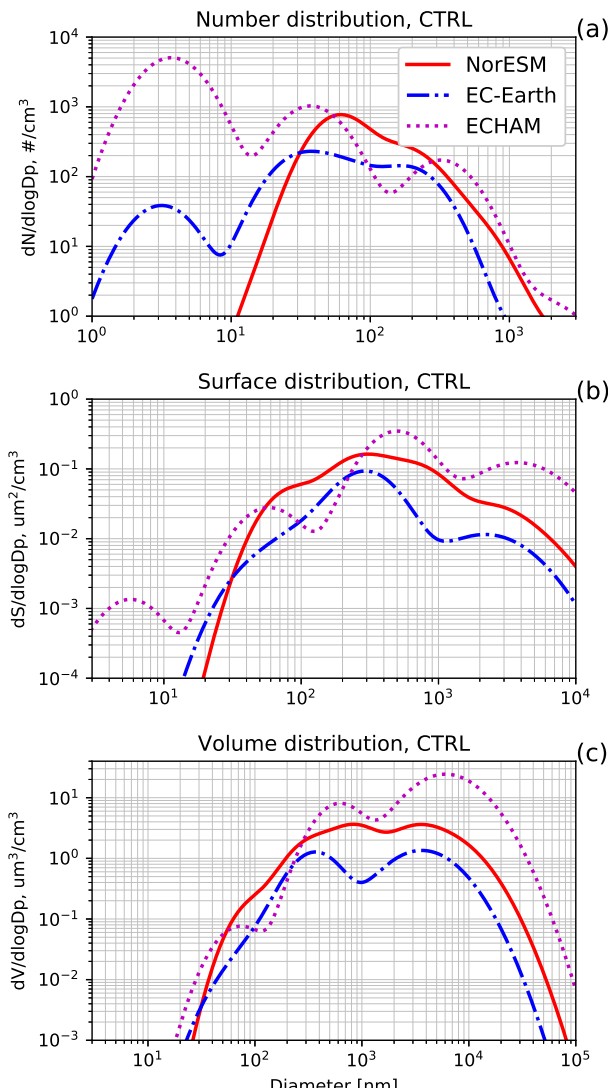

**Figure 3.** Globally averaged aerosol number size distributions (a), surface size distributions (b) and volume size distributions from the three models from the *CTRL* simulation. The diameters are the dry diameters. Note the different scales on the x-axis in the subplots.

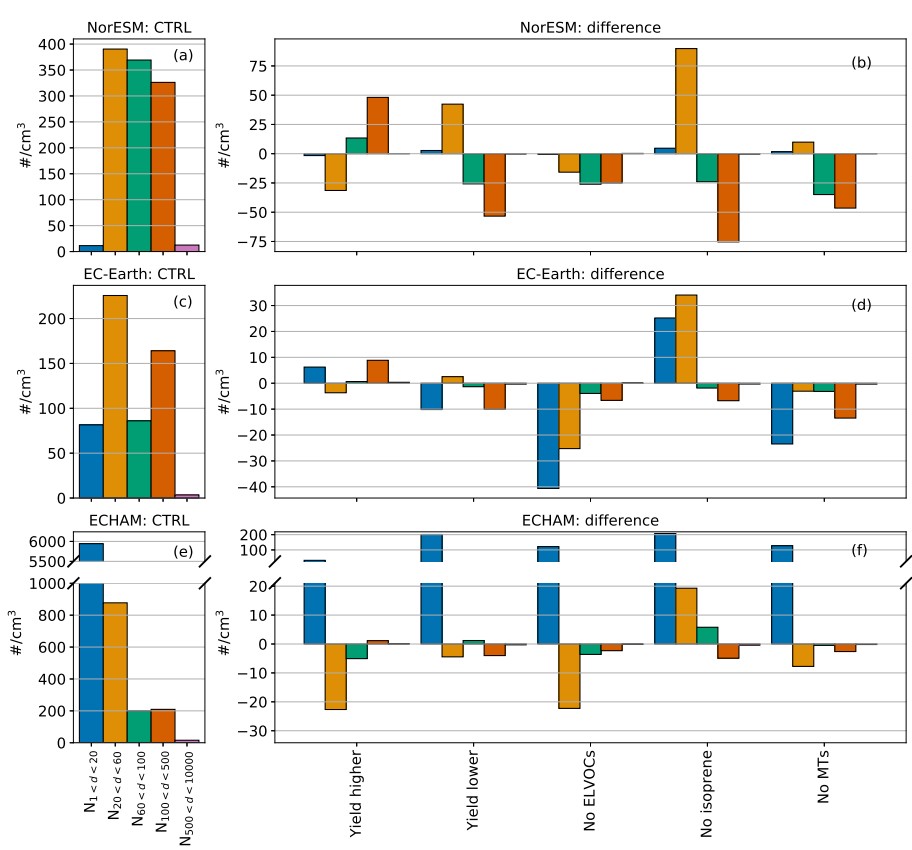

**Figure 4.** Bar plots of the number concentration of particles of selected sizes. In a, c and e the number concentrations from the *CTRL* simulations are shown. In b, d and f the absolute differences between the sensitivity simulations and the control simulations are shown for each size bin. Note that there are different scales used for the different models.



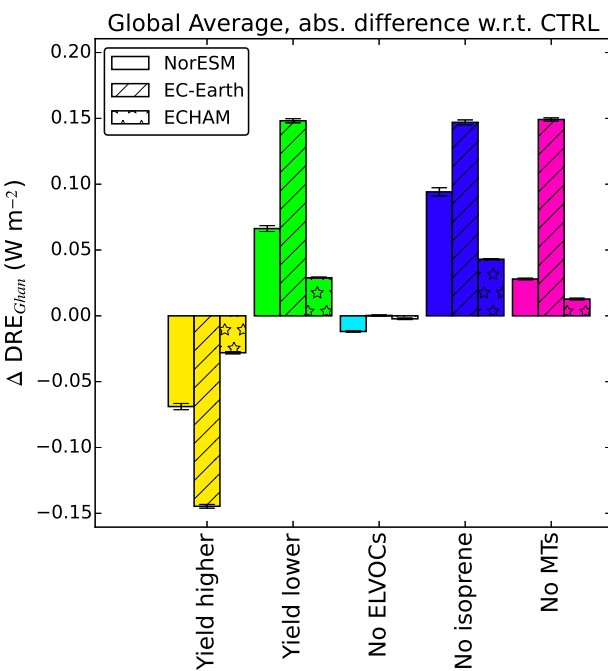

**Figure 5.** Bar plot of the average direct radiative effect (DRE$_{Ghan}$) difference between each sensitivity simulation and the *CTRL* simulations. Coloured bars indicate a significant difference in the simulation averages with a 95% confidence interval. The error bars denote the standard error of mean of the yearly differences.



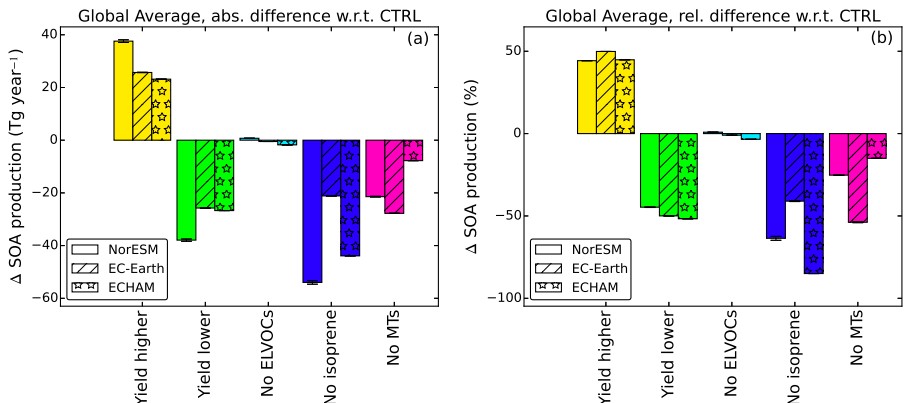

**Figure 6.** Bar plot of the average yearly SOA production changes in the three models. In (a), the absolute difference between each sensitivity simulation and the *CTRL* simulations are shown and in (b) the relative differences between the sensitivity simulations and the control simulations are shown for all three models. Coloured bars indicate a significant difference in the simulation averages with a 95% confidence interval. The error bars denote the standard error of mean of the yearly differences.



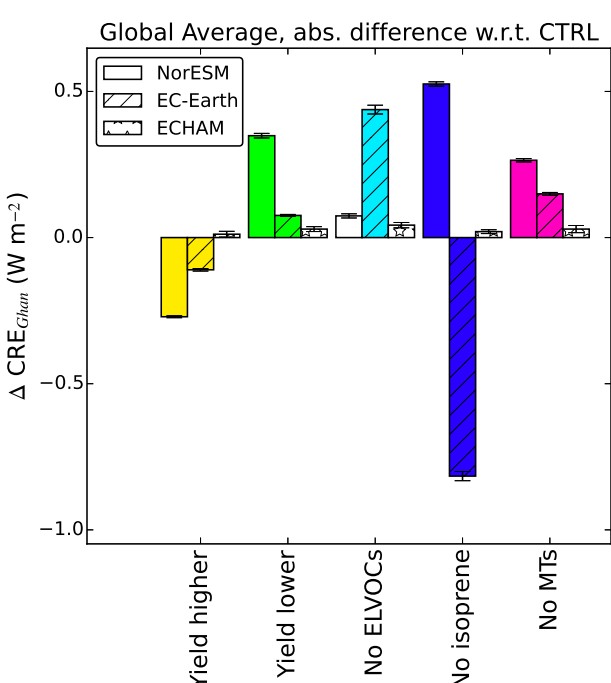

**Figure 7.** Bar plot of the global average cloud radiative effect (CRE$_{Ghan}$) between each sensitivity simulation and the *CTRL* simulations. Coloured bars indicate a significant difference in the simulation averages with a 95% confidence interval. The error bars denote the standard error of mean of the yearly differences.

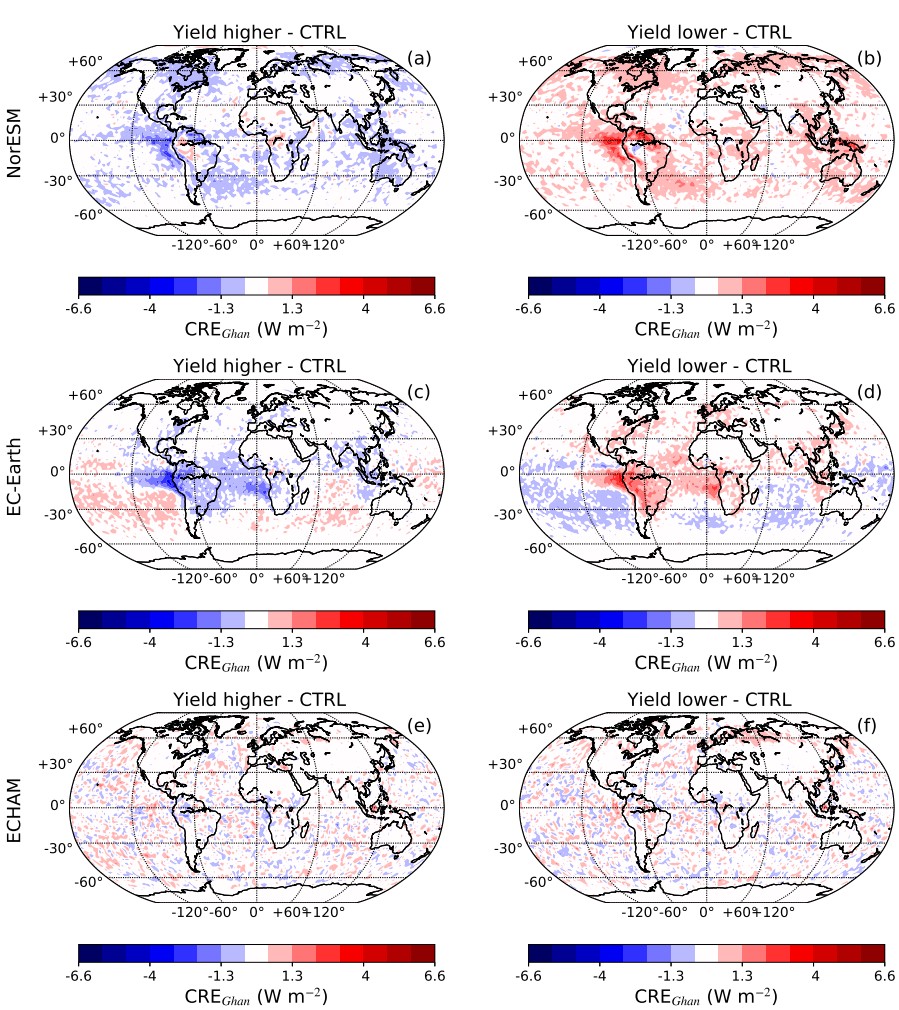

**Figure 8.** Maps of the difference in the average cloud radiative effect (CRE$_{Ghan}$) between the *Yield higher* (a,c and d) and *Yield lower* (b, d and h) with respect to the *CTRL simulation*. This is shown for NorESM (a and b), EC-Earth (c and d) and ECHAM (e and f).



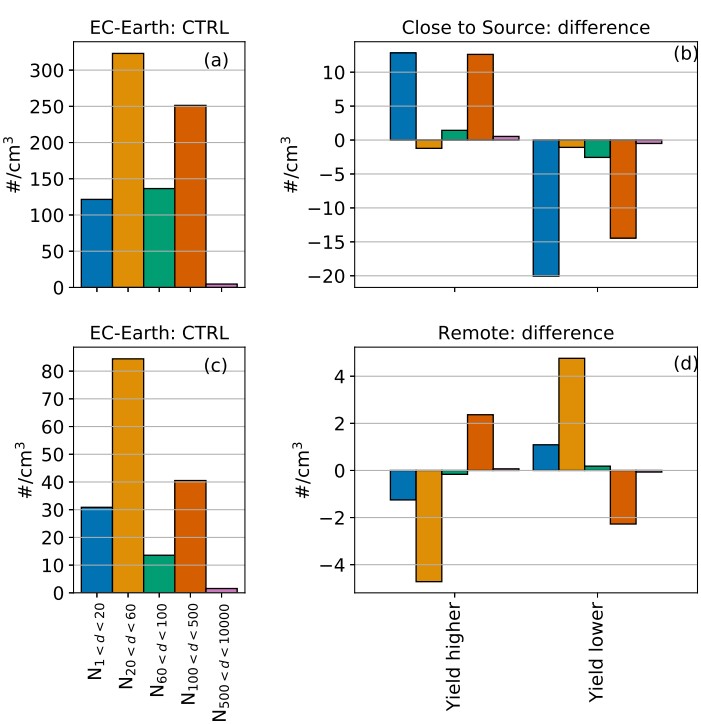

**Figure 9.** Bar plots of the number concentration of particles of selected sizes ranges for EC-Earth, close to and far away from the sources. In a and c the number concentrations from the *CTRL* simulations are shown. In b and d the differences between the sensitivity simulations and the *CTRL* simulations are shown for each size bin. Note that there are different scales used for the different regions. The areas defined as close and remote are based on changes in CCN concentrations (positive or negative) at 1 % supersaturation. These areas can be seen in Fig. S21.



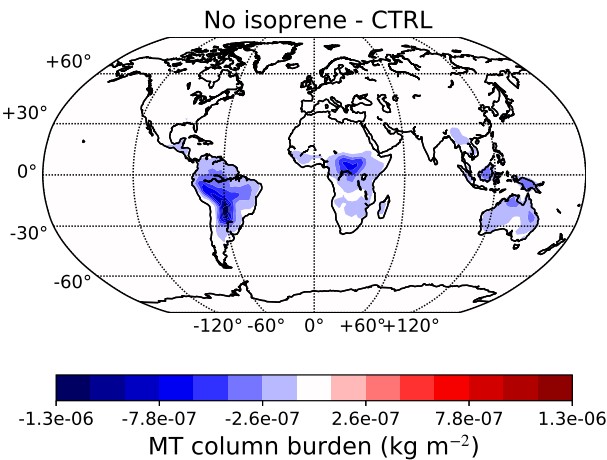

**Figure 10.** Map of the difference in the average MT column burden between the *No isoprene* and *CTRL* simulation for EC-Earth.



**Table 1.** This Table lists similarities and dissimilarities between the three ESM of particularly relevance for this study.

| Properties | NorESM | EC-Earth | ECHAM |
|---|---|---|---|
| Aerosol model | OsloAero | M7 | M7 |
| BVOC emissions | MEGAN v2.1 interactive | MEGAN-MACC prescribed | MEGAN-MACC prescribed |
| Oxidant fields | Prescribed | Interactive | Prescribed |
| Oxidised BVOC Tracers | ELVOC / L/SVOC | ELVOC / L/SVOC | ELVOC / L/SVOC |
| ELVOC formed from | MTs | isoprene and MTs | isoprene and MTs |
| Binary nucl. param. | Vehkamäki et al. (2002) | Vehkamäki et al. (2002) | Vehkamäki et al. (2002) |
| BL nucleation rate J = | $A_1[H_2SO_4] + A_2[ELVOC]$ | $A_3[H_2SO_4]^2 \times [ELVOC]$ | $A_1[H_2SO_4] + A_2[ELVOC]$ |
| Cloud activation scheme | Abdul-Razzak and Ghan (2000) | Abdul-Razzak and Ghan (2000) | Abdul-Razzak and Ghan (2000) |

$A_1 = 6.1 \times 10^{-7} \ s^{-1}$
$A_2 = 3.9 \times 10^{-8} \ s^{-1}$
$A_3 = 3.27 \times 10^{-21} \ cm^{-6} \ s^{-1}$



**Table 2.** Yields and resulting products from the reactions of BVOCs with oxidants to form ELVOC and L/SVOC for the three ESMs. In ECHAM, there are different yields for endocyclic MTs and the other MTs. The equations for the endocyclic MTs are written separately and the other MTs are shown on the same rows as the MTs in the other models.

| Model | NorESM | EC-Earth | ECHAM |
|---|---|---|---|
| Isop. + OH | 0.05 L/SVOC | 0.0097 L/SVOC + 0.0003 ELVOC | 0.0482 L/SVOC + 0.0018 ELVOC |
| Isop. + $O_3$ | 0.05 L/SVOC | 0.0099 L/SVOC + 0.0001 ELVOC | 0.0498 L/SVOC + 0.00016 ELVOC |
| Isop. + $NO_3$ | 0.05 L/SVOC | - | - |
| MTs + OH | 0.15 L/SVOC | 0.14 L/SVOC + 0.01 ELVOC | 0.14 L/SVOC + 0.01 ELVOC |
| MTs + $O_3$ | 0.15 ELVOC | 0.1 L/SVOC + 0.05 ELVOC | 0.147 L/SVOC + 0.003 ELVOC |
| MTs + $NO_3$ | 0.15 L/SVOC | - | - |
| Endocyclic MTs + OH | - | - | 0.145 L/SVOC + 0.005 ELVOC |
| Endocyclic MTs + $O_3$ | - | - | 0.1 L/SVOC + 0.05 ELVOC |