# Peer review of "Large difference in aerosol radiative effects from BVOC-SOA treatment in three ESMs"

_Atmospheric Chemistry and Physics, 2019_

## Referee Comment (RC1) · Anonymous Referee #3 · 1 Mar 2020

This paper compares the responses of three Earth System Models to changes in BVOC emissions and BSOA yields, with regard to radiative effects. It is very useful to compare models in this way, and important to show that such models can give very different results if used to investigate the climate impacts of BVOCs.

Although the paper is generally well written, I think the caveats and conclusions need to reflect some of the issues which this study couldn't address. I actually disagree with the last line of the conclusions which says that "in particular" one needs to work more on how NPF parameterizations affect size distributions. Although I agree that these NPF issues are important, I don't see the evidence that this is the main problem with SOA modeling. I suspect it reflects more the author's plans and interests than the general level of SOA understanding.

[Figure]

A few connected issues are:

The authors clearly state that their semi-volatile SOA compounds aren't semi-volatile - they are formed as inert and condensed species upon BVOC oxidation. This is a major assumption, but the implications are not discussed anywhere in the manuscript. Compared to a true VBS equilibrium framework these models presumably give much more SOA in less polluted regions, and in the free troposphere. Isn't this one of the main uncertainties of all of these models?

No model evaluation is presented, and no indication is given as to where such evaluations can be found. The model versions used here seem to differ from those used by Tsigaridis et al, 2014, so readers have no idea if the modelled SOA are reasonable or not.

There is no comparison of the BSOA production calculated here with that of other studies, e.g. Hallquist et al 2009 or Tsigaridis et al. 2014.

Although the paper mentions the Spracklen et al 2011 study concerning anthropogenic influence (also commented below), there is no mention of the role of NOx on the BSOA yields assumed in this paper. Most VBS schemes would have both high and low NOx yields, and perform some interpolation between them depending on oxidant availability. If one believes in some anthropogenic influence, then the assumed yields should depend on NOx as well as oxidants.

I also missed any mention of POA, BBOA, or ASOA in Sect. 2 and elsewhere. Do these models only have BSOA? What are the implications of this?

Other issues

p1, L10. Why 10 years? The importance of BVOC to SOA formation has been known for decades!

p2, L1. Say implies rather than introduces

p2, L5. Not all BVOC are "quickly" oxidized.

p2, L10. Give reference for the direct aerosol effect comment

p2, L11. Add radiation as one of the major drivers of BVOC emissions

p2, L12. It is usually good to cite articles if possible, and could have used e.g. Hantson et al 2017 or Schurgers et al. 2009 here.

p2, L13. There are several studies suggesting that increased $CO_2$ can inhibit BVOC emissions (e.g. Arneth et al, 2007, refs in Hantson et al 2017). This is also a major source of uncertainty that needs a mention.

p2, L22. Season matters. BSOA often dominates SOA in summertime, but there is plenty of evidence that in wintertime wood-burning often dominates or plays a major role (e.g. Brown et al., 2016, Glasius et al., 2018).

p2, L24. Although the Spracklen et al 2011 study was very innovative and interesting, there are several issues with the conclusions, see e.g. Hodzic & Jimenez, 2011. This question is very complex and unresolved as far as I know.

p3, L30-33. I think the sentences starting on L30 are very clear, fair, and with the important caveat represented by the last sentence. These lines could usefully be used in the abstract.

p5, L2. The descriptions are brief, not "detailed".

p5, L13. Which MEGAN version?

p6, L1. So, is this a new version of NorESM, or a version used just for this study? If the latter, then the conclusions aren't relevant to other NorESM work, which would seem to remove some of the point of including this model.

p6, L9. I think you mean hydroxyl, not hydroxide, and better to say nitrate radical, as nitrate is often used for the aerosol compound.

p6, L11. This was confusing. If I understand right, one has two types of ELVOC then, one that can influence NPF, and the other behaves exactly as L/SVOC. As you ELVOC and L/SVOC compounds have the same mass, why not simply put the non-NPF ELVOC in as L/SVOC?

p7, L32. This was also confusing. Table 1 suggests that ECHAM uses fixed yields of L/SVOC and ELVOC, but L32 suggests partitioning depends on pre-existing organic mass. And what is meant by pre-existing OM? Does this influence the DRE/CRE calculations?

p9, L2-7. I am not sure the argument about interactive oxidants can explain a factor of 3. Sure, when the BVOC are emitted one can expect reduced OH and other oxidants, but this just delays the oxidation close to the surface. Isoprene which isn't oxidized near the surface will still be oxidized a little further up in the troposphere. Why would the total amount change by a factor of 3? Did you check changes in oxidant fields associated with this argument?

p16-17. This is where I think the limitations and results of this study need to be put alongside the many other uncertainties surrounding SOA modelling.

p26, Fig. 1. State which years are shown here. Also, I was surprised to see no error-bars on the ECHAM runs, and that even those for NorESM were so small. Did ECHAM also just use one fixed year (2000) of BVOC emission, same as EC-Earth?

References

Arneth, A. et al, 2007, CO2 inhibition of global terrestrial isoprene emissions, potential implications for atmospheric chemistry. Geophys. Res. Lett. 34 http,//dx.doi.org/10.1029/2007gl030615

Brown, SG et al., Wintertime Residential Biomass Burning in Las Vegas, Nevada; Marker Components and Apportionment Methods Atmosphere, 2016, 7, 58

Glasius, M. et al., Composition and sources of carbonaceous aerosols in Northern

[Figure]

Europe during winter Atmospheric Environment, 2018, 173, 127-141

Hodzic, A. & Jimenez, J. L. Modeling anthropogenically controlled secondary organic aerosols in a megacity, a simplified framework for global and climate models Geoscientific Model Dev., 2011, 4, 901-917

Schurgers, G. et al., European emissions of isoprene and monoterpenes from the Last Glacial Maximum to present Biogeosciences, 2009, 6, 2779-2797

---

## Referee Comment (RC2) · Anonymous Referee #1 · 31 Mar 2020

This is a nice modeling study that provides key insights about how three different global climate models respond differently in terms of aerosol-cloud-radiation interactions with changes in VOC emissions, NPF etc. I have following suggestions for clarifications and discussions that will further improve the interpretability and take-home messages. One of the key points to be discussed is how different model responses to perturbations in VOC emissions, NPF etc. in different regions can be evaluated with observations. These observations could be a combination of surface-, aircraft and satellite observations. Comparison to observed responses of SOA, CCN, CDNC etc. to perturbations are critical for reducing model uncertainties.

Line 360: Why does increase in accumulation mode particles over remote regions cause a decrease in smaller (20-60 nm) particles in this model? If SOA gas precur-

sors are transported to remote regions they should be effective in growing 20-60 nm particles to CCN sizes. This should increase CCN, CDNC and CRE. But it seems the opposing effects on accumulation and Aitken particles is related to transport of already formed accumulation mode SOA (upstream of the remote regions) that decreases smaller (20-60 nm) particles due to coagulation?

Conceptually, if in EC-Earth the timescale for SOA formation could increase by reducing oxidants over remote regions (due to changes in gas-phase chemistry), this should increase CCN?

Line 395: The strong reduction in nucleation rates in EC_Earth over large anthropogenic regions (due to shutting off ELVOCs) seems less realistic. BL nucleation rates should not be shut off when ELVOCs are zero. Observational evidence suggests that presence of $H_2SO_4$ will nucleate particles while low volatility organics grow these particles to CCN sizes (e.g. Riipinen et al. 2011 cited in this paper). Please comment on how realistic the nucleation rates are in EC_Earth based on observational evidence.

Line 425: Why does increase in OH in EC_Earth (in no isoprene) cause reduction in $O_3$?

Line 425: Why does increase in oxidants not affect amount of SOA formed? If more SOA is formed close to sources it can undergo wet removal. Also dry deposition of SOA precursors would reduce available precursors. This could also change if SOA precursors L/SVOCs and their multigenerational chemistry are represented by a volatility basis set framework. For example see: https://agupubs.onlinelibrary.wiley.com/doi/10.1002/2014JD022563

Line 495-505: It is clear that the three global models would produce different results if used to investigate climate impacts of BVOCs. But observations could provide insights about changes in model behavior with respect to changes in VOC emissions, NPF, presence/absence of ELVOCs and $H_2SO_4$. Also one of the key insights from this work is importance of interactive oxidants. Future model measurement comparisons could

focus on evaluating model responses to perturbations in VOC emissions, NPF, SO2, NOx emissions etc. with respect to observations for e.g. in cleaner locations such as over the Amazon: https://www.nature.com/articles/s41467-019-08909-4

Minor clarification comments: Page 2: Shrivastava et al. 2015 predicted biomass burning to be the largest SOA source followed by biogenic SOA. Fossil-fuel SOA burden was the smallest and contributed less than one-third of biogenic SOA (see their Figure 5). Please rephrase.

Sum of isoprene, monoterpene and sesquiterpene emissions to BVOCs should sum to 100%. Right now 50+15+3 is 68%. Am I missing something here?

Line 270-275: Why does ECHAM have more particles in nucleation mode than EC-Earth? Does EC_Earth treat nucleation mode? I would expect since NorESM has no nucleation mode, it would have the lowest particle numbers since the SOA mass goes to larger particles, but seems EC_Earth has the lowest numbers. Why?

Line 285: I would think if aerosols are located above bright/reflective surfaces, they will absorb more sunlight and contribute more strongly to DRE. But it seems in NOrESM when there is higher AOD above bright surfaces it results in lower DRE forcing. Above deserts, dust particles should also be light absorbing. So this statement is confusing. Please explain.

Also how do the various models determine fraction going to soluble versus insoluble modes?

Is hygroscopicity of insoluble mode assumed to be zero?

What is assumed hygroscopicity of SOA in soluble/insoluble modes and in different models? I would expect hygroscopicity of SOA to play a stronger role in SOA dominated regions like the Amazon. Please comment on the role of hygroscopicity and water uptake on aerosol-cloud-radiation interactions.

Why does EC-Earth experience particle number changes farther downstream related

to other models? Is it related to interactive oxidants in this model that reduce SOA formation close to sources but increase it downwind relative to a model with prescribed oxidants?

Line 335: It says "smaller" CWP due to higher CCN, CDNC and smaller effective radius. I think the authors meant "larger" CWP?

---

## Referee Comment (RC3) · Anonymous Referee #2 · 1 Apr 2020

This manuscript investigates the differences in the radiative effects induced by biogenic secondary organic aerosol (SOA) as computed by three different ESMs. It points as drivers of these differences, in particular, to the parameterizations of new particle formation, biogenic SOA treatment and biogenic volatile organic compounds emissions changes by climate-feedbacks and by land-use changes. The manuscript is nicely written and suitable for publication in ACP after further improvements as detailed by the two other reviewers. I have some comments in addition to the long list already provided by the two other reviewers.

I suggest the authors discuss/compare not only with the Tsigaridis et al (2014) paper on organic aerosols but also with the Fanourgakis et al (2019) on CCN, where several models with different treatment of NPF and of OA have been compared. How the SOA

budget calculated in this study compare with the earlier estimates reported in both the above mentioned papers. How the aerosol number concentrations and the CCN calculated compare with Fanourgakis et al. What do we learn from comparison with these earlier studies?

Furthermore, I would like to see some conclusive discussion on the importance of the interactive calculations of the oxidants (only EC-Earth having this feature) and shows very different behavior compared to the other two ESMs for the simulations without biogenic VOCs, pointing to potentially some chemical feedback mechanisms though oxidants impact on SOA formation that is not discussed.

Extra specific comments:

1- Line 45: Kanakidou et al. (2000) and Carlton et al (2008) have discussed the anthropogenic control on BSOA before Spracklen et al. (2011) I think they deserve citation here.

2- Lines 60-62: Assumption of irreversible condensation of organics is a working assumption for ELVOC but not for SVOC. I think this has to be discussed here since for SVOC this will lead to an overestimate of SOA formation.

3- Lines 129 & Table 1: The model description leaves the impression to the reader that MEGAN is used on line in all three ESMs while in Table 1 it is stated that two among the three ESMs use recalculated fields. This has to be clarified.

Ref:

Carlton et al., To What Extent Can Biogenic SOA be Controlled? Environ. Sci. Technol. 2010, 44, 3376–3380

Fanourgakis et al. Evaluation of global simulations of aerosol particle and cloud condensation nuclei number, with implications for cloud droplet formation Atmos. Chem. Phys., 19, 8591–8617, 2019, https://doi.org/10.5194/acp-19-8591-2019

Kanakidou et al., Human-activity-enhanced formation of organic aerosols from biogenic hydrocarbon oxidation, JGR, 105,D7,9243-9254, 2000

Tsigaridis et al. The AeroCom evaluation and intercomparison of organic aerosol in global models Atmos. Chem. Phys., 14, 10845–10895, 2014, www.atmos-chem-phys.net/14/10845/2014

———————————————

---

## Author Comment (AC1) · 12 May 2020

**Authors responses**

Moa K. Sporre[1,2], Sara M. Blichner[1], Roland Schrödner[3], Inger H. H. Karset[1], Terje K. Berntsen[1,4], Twan van Noije[5], Tommi Bergman[5,6], Declan O'Donnell[6], and Risto Makkonen[6,7]

[1]Department of Geosciences, University of Oslo, Postboks 1022 Blindern, 0315 Oslo, Norway
[2]Now at: Department of Physics, Lund University, Box 118, 22100 Lund, Sweden
[2]Institute for Tropospheric Research, Permoserstr. 15, 04318 Leipzig, Germany
[4]CICERO Center for International Climate Research, Postboks 1129 Blindern, 0318 Oslo, Norway
[5]Royal Netherlands Meteorological Institute (KNMI), PO Box 201, 3730 AE De Bilt, the Netherlands
[6]Climate System Research, Finnish Meteorological Institute, P.O. Box 503, FI-00101, Helsinki, Finland
[7]Institute for Atmospheric and Earth System Research / Physics, Faculty of Science, University of Helsinki, P.O. Box 64, FI-00014, Finland

We would like to thank the reviewers for their comments which will improve this manuscript. We present our responses to the questions and comments below. The comments from the reviewer are written first and our responses follow in italic. We will answer the comments from all three referees in this document.

5 **Reviewer number 1**

Line 360: Why does increase in accumulation mode particles over remote regions cause a decrease in smaller (20-60 nm) particles in this model? If SOA gas precursors are transported to remote regions they should be effective in growing 20-60 nm particles to CCN sizes. This should increase CCN, CDNC and CRE. But it seems the opposing effects on accumulation and Aitken particles is related to transport of already formed accumulation mode SOA (upstream of the remote regions) that

10 decreases smaller (20-60 nm) particles due to coagulation?

*We agree with the reviewer that transport of already formed accumulation mode particles is a plausible explanation for the change in the size distributions over the remote regions. With the model output we have we can however not confirm that this is the explanation.*

15 Conceptually, if in EC-Earth the timescale for SOA formation could increase by reducing oxidants over remote regions (due to changes in gas-phase chemistry), this should increase CCN?

*Yes. The effect has also been seen in other models (e.g. Karset et al., 2018).*

Line 395: The strong reduction in nucleation rates in EC-Earth over large anthropogenic regions (due to shutting off

20 ELVOCs) seems less realistic. BL nucleation rates should not be shut off when ELVOCs are zero. Observational evidence suggests that presence of H2SO4 will nucleate particles while low volatility organics grow these particles to CCN sizes (e.g. Riipinen et al. 2011 cited in this paper). Please comment on how realistic the nucleation rates are in EC-Earth based on observational evidence.

*We agree with the reviewer that completely shutting off the boundary layer nucleation when ELVOCs are zero is not a very realistic. However, the complete absence of ELVOCs is a sensitivity test and not a very likely scenario. The nucleation rate parameterization in EC-Earth is based on the experiments from the CLOUD chamber and has performed well in other models when compared to measurements (Riccobono et al., 2014). However, the nucleation rates in EC-Earth may be underestimated over oceans since there are no VOC emissions from the ocean in this model in these simulations.*

Line 425: Why does increase in OH in EC-Earth (in no isoprene) cause reduction in O3?

*Since $O_3$ is produced when isoprene is oxidised. We have changed this text to make this clear and it now reads: "The concentration of $O_3$ is reduced since the oxidation of of isoprene results in the production of $O_3$. However, the loss rate of MT to $O_3$ oxidation is less important than OH, and thus the overall result is a reduction in column burden."*

Line 425: Why does increase in oxidants not affect amount of SOA formed? If more SOA is formed close to sources it can undergo wet removal. Also dry deposition of SOA precursors would reduce available precursors. This could also change if SOA precursors L/SVOCs and their multigenerational chemistry are represented by a volatility basis set framework. For example see: https://agupubs.onlinelibrary.wiley.com/doi/10.1002/2014JD022563.

*The ELVOC and L/SVOCs in these three models do not undergo any wet or dry deposition. Therefore, all formed ELVOC and L/SVOCs will form SOA eventually. How close to the sources this will occur depends on the available oxidants in EC-Earth. We agree with the reviewer that this would not occur if a volatility basis set framework would have been used in the models.*

Line 495-505: It is clear that the three global models would produce different results if used to investigate climate impacts of BVOCs. But observations could provide insights about changes in model behavior with respect to changes in VOC emissions, NPF, presence/absence of ELVOCs and H2SO4. Also one of the key insights from this work is importance of interactive oxidants. Future model measurement comparisons could focus on evaluating model responses to perturbations in VOC emissions, NPF, SO2, NOx emissions etc. with respect to observations for e.g. in cleaner locations such as over the Amazon: https://www.nature.com/articles/s41467-019-08909-4

*We agree that comparing the models to measurements is important, though out of the scope of this manuscript. A comment regarding model evaluation has been added to the further discussions section.*

Minor clarification comments: Page 2: Shrivastava et al. 2015 predicted biomass burning to be the largest SOA source followed by biogenic SOA. Fossil-fuel SOA burden was the smallest and contributed less than one-third of biogenic SOA (see their Figure 5). Please rephrase.

*Yes, this has been corrected.*

Sum of isoprene, monoterpene and sesquiterpene emissions to BVOCs should sum to 100%. Right now 50+15+3 is 68%. Am I missing something here?

*The rest of the emitted BVOC compounds (32 %) are not as relevant for aerosol formation (e.g. methanol, acetone, CO etc). We*
60  *have changed the text to make this clear. The sentence now reads: "However, in ESMs these are often reduced to be represented*
*by 2-3 tracers, usually isoprene, monoterpenes (MTs) and sesquiterpenes which constitute the main contributors to aerosol*
*formation and are estimated to constitute around 50%, 15% and 3% respectively of the total BVOC emissions [...]"*

Line 270-275: Why does ECHAM have more particles in nucleation mode than EC-Earth? Does EC-Earth treat nucleation
65  mode? I would expect since NorESM has no nucleation mode, it would have the lowest particle numbers since the SOA mass
goes to larger particles, but seems EC-Earth has the lowest numbers. Why?
*One of the reasons why ECHAM has more particles in the nucleation mode is the diameter at which the newly formed particles*
*are introduced to this mode. This diameter is lower than the nucleation mode median in ECHAM which means that the mode*
*median is reduced when there is new particle formation. This reduces the amount of particles that are transferred to the Aitken*
70  *mode which means more particles are left in the nucleation mode in ECHAM. This does not occur in EC-Earth because here the*
*diameter of the newly formed particles is above the mode median. We write about this in the discussion. Another explanation*
*for the large difference between the nucleation modes in EC-Earth and ECHAM could be that the nucleation rates in ECHAM*
*might be overestimated. Moreover, the nucleation rates over the oceans in EC-Earth may be underestimated as described in an*
*earlier comment.*
75  *There are most likely many different reasons to why EC-Earth has lower particle concentrations than NorESM. One possible*
*explanation could be higher wet deposition in EC-Earth due to a 6 hour coupling time step between TM5 and IFS. This means*
*that it can rain for 6 hours in a row in TM5. The difference in the aerosol size distributions can also be linked to how the*
*emissions (which are mass based) are partitioned into the different size modes, which is different in the models.*
*We have added the following sentences to the manuscript to address these questions: "There are likely many different explana-*
80  *tions to why the size distributions and aerosol number concentrations are different in the models. Some plausible explanations*
*include differences in wet deposition, nucleation rates and how the emissions in general are partitioned into the aerosol modes."*

Line 285: I would think if aerosols are located above bright/reflective surfaces, they will absorb more sunlight and contribute
more strongly to DRE. But it seems in NorESM when there is higher AOD above bright surfaces it results in lower DRE
85  forcing. Above deserts, dust particles should also be light absorbing. So this statement is confusing.
Please explain.
*The high AOD over deserts in NorESM is mainly a result of high desert dust emissions. These aerosols are mainly reflecting*
*incoming solar radiation and not absorbing very much. Reflective aerosols above a light surface (such as deserts) have almost*
*the same properties as the surface below and therefore only have a modest radiative effect. We have added a sentence to the*
90  *manuscript explaining that the high AOD over desserts in NorESM is associated with large dust emissions.*

Also how do the various models determine fraction going to soluble versus insoluble modes?
*Emissions are partitioned between soluble/insoluble modes in fixed proportions. E.g. in TM5 it is assumed that all carbona-*

ceous aerosol from sources other than biofuel burning and open biomass burning to be emitted as insoluble (into the Aitken

95 mode). Insoluble particles are aged through condensation of sulfuric acid, and when coated with $H_2SO_4$, are transferred to the corresponding soluble size mode. Coagulation of insoluble and soluble particles also results in soluble particles.

Is hygroscopicity of insoluble mode assumed to be zero?

*Yes, in both EC-Earth and ECHAM.*

100

What is assumed hygroscopicity of SOA in soluble/insoluble modes and in different models? I would expect hygroscopicity of SOA to play a stronger role in SOA dominated regions like the Amazon. Please comment on the role of hygroscopicity and water uptake on aerosol-cloud-radiation interactions.

*The hygroscopicity parameter for organic aerosols (including SOA) in NorESM is 0.14 (Kirkevåg et al., 2018). The kappa for*

105 *soluble organic aerosols in EC-Earth in terms of cloud activation is 0.1 and the kappa for organics in ECHAM is 0.06. For the insoluble mode see the previous question. There is a sentence addressing hygroscospicity included in Section 3.2.2. of the manuscript.*

Why does EC-Earth experience particle number changes farther downstream related to other models? Is it related to inter-

110 active oxidants in this model that reduce SOA formation close to sources but increase it downwind relative to a model with prescribed oxidants?

*Yes, we consider this a plausible explanation. We have added a paragraph to the discussions section where we discuss the interactive oxidants a bit more extensively.*

115 Line 335: It says "smaller" CWP due to higher CCN, CDNC and smaller effective radius. I think the authors meant "larger" CWP?

*This is correct. We have changed "smaller" to "larger" in the text.*

**Reviewer number 2**

120 I suggest the authors discuss/compare not only with the Tsigaridis et al (2014) paper on organic aerosols but also with the Fanourgakis et al (2019) on CCN, where several models with different treatment of NPF and of OA have been compared. How the SOA budget calculated in this study compare with the earlier estimates reported in both the above mentioned papers. How the aerosol number concentrations and the CCN calculated compare with Fanourgakis et al. What do we learn from comparison with these earlier studies?

125 *We have added a comparison to the Tsigaridis et al. (2014) study. We have read the Fanourgakis et al (2019) very carefully and though it is a very interesting paper we find it hard to make comparison between the results from that study and this manuscript. In terms of SOA budget, the Fanourgakis et al (2019) study uses OA concentrations which is not a parameter that we have investigated in this study. Moreover, we have not included results regarding CCN concentrations in this study and*

can therefore not compare these concentrations to those in Fanourgakis et al (2019). Both studies include aerosol number concentrations above 3 nm but Fanourgakis et al (2019) uses near-surface concentrations while we used average values at pressure above 850 hPa. Since the concentrations of particles changes rapidly in the lowermost layers of the models we do not see it fit to compare these concentrations either. However, studies where models are compared to measurements are of great importance in developing models and we have added a comment to emphasize this in the discussion.

Furthermore, I would like to see some conclusive discussion on the importance of the interactive calculations of the oxidants (only EC-Earth having this feature) and shows very different behavior compared to the other two ESMs for the simulations without biogenic VOCs, pointing to potentially some chemical feedback mechanisms though oxidants impact on SOA formation that is not discussed. *We have added a paragraph to the Further discussions section where we discuss the interactive oxidants in EC-Earth further.*

Extra specific comments: 1- Line 45: Kanakidou et al. (2000) and Carlton et al (2008) have discussed the anthropogenic control on BSOA before Spracklen et al. (2011) I think they deserve citation here. *We have added these two references here.*

2- Lines 60-62: Assumption of irreversible condensation of organics is a working assumption for ELVOC but not for SVOC. I think this has to be discussed here since for SVOC this will lead to an overestimate of SOA formation. *We have added a paragraph to the discussion where we include a discussion on the irreversible condensation of SVOCs.*

3- Lines 129 Table 1: The model description leaves the impression to the reader that MEGAN is used on line in all three ESMs while in Table 1 it is stated that two among the three ESMs use recalculated fields. This has to be clarified. *MEGAN has been used to calculate the emissions for all three models. However, for two of the models this had been done offline and these are denoted prescribed in the table. We have rewritten the text in the methods section such that it now reads: "Moreover, all three ESMs use the Model of Emissions of Gases and Aerosols from Nature (MEGAN) version 2.1 (Guenther et al., 2012). However, in NorESM, MEGAN is run interactively while the other two models use emissions produced in offline simulations with MEGAN."*

**Reviewer number 3**

Although the paper is generally well written, I think the caveats and conclusions need to reflect some of the issues which this study couldn't address. I actually disagree with the last line of the conclusions which says that "in particular" one needs to work more on how NPF parameterizations affect size distributions. Although I agree that these NPF issues are important, I don't see the evidence that this is the main problem with SOA modeling. I suspect it reflects more the author's plans and interests than the general level of SOA understanding.

*We have added a paragraph in the further discussions section where we address limitations of these models. This investigation does not investigate how important these NPF issues are in comparison to other uncertainties regarding SOA and we do not*

*make any such conclusions. Our view is that the NPF issues is not competing with other SOA modelling issues but is rather a part of the overall uncertainty regarding SOA impact on climate. Nevertheless, we have removed the words "in particular" from the last sentence.*

A few connected issues are:

The authors clearly state that their semi-volatile SOA compounds aren't semi-volatile - they are formed as inert and condensed species upon BVOC oxidation. This is a major assumption, but the implications are not discussed anywhere in the manuscript. Compared to a true VBS equilibrium framework these models presumably give much more SOA in less polluted regions, and in the free troposphere. Isn't this one of the main uncertainties of all of these models?

*We have added a paragraph to the discussion where we include a discussion on the irreversible condensation of SVOCs and the VBS equilibrium framework.*

No model evaluation is presented, and no indication is given as to where such evaluations can be found. The model versions used here seem to differ from those used by Tsigaridis et al, 2014, so readers have no idea if the modelled SOA are reasonable or not.

*We have added a sentence comparing the modelled SOA in this study to the results in Tsigaridis et al. (2014): "The produced SOA mass in the models are in the range of the values found in Tsigaridis et al. (2014) but are higher than the median of the models included in that study." However, a large model evaluation is beyond the scope of this manuscript.*

There is no comparison of the BSOA production calculated here with that of other studies, e.g. Hallquist et al 2009 or Tsigaridis et al. 2014.

*The BSOA is included in the SOA production terms but we have not calculated this fraction explicitly in this study. Therefore we have not included a comparison of the BSOA separately in this paper.*

Although the paper mentions the Spracklen et al 2011 study concerning anthropogenic influence (also commented below), there is no mention of the role of NOx on the BSOA yields assumed in this paper. Most VBS schemes would have both high and low NOx yields, and perform some interpolation between them depending on oxidant availability. If one believes in some anthropogenic influence, then the assumed yields should depend on NOx as well as oxidants.

*The models used in this study have no yield dependence on NOx or oxidants. We have added a paragraph discussing SOA and the limitations of these models in terms of their SOA parametrisations to the discussion section.*

I also missed any mention of POA, BBOA, or ASOA in Sect. 2 and elsewhere. Do these models only have BSOA? What are the implications of this?

*We have added a paragraph discussing SOA and the limitations of these models in terms of their SOA parametrisations to the*

*discussion section. Here we discuss POA and ASOA. We do not know what BBOA is but we assume that the reviewer means*
200 *BPOA, which is also discussed in the new paragraph.*

Other issues

p1, L10. Why 10 years? The importance of BVOC to SOA formation has been known for decades!
205 *We have removed "past 10". The sentence now reads: "As the understanding of the importance of BVOCs for aerosol formation has increased over the years these processes have made their way into Earth System Models (ESMs)"*

p2, L1. Say implies rather than introduces
*We have changed this according to the reviewer's comment.*
210

p2, L5. Not all BVOC are "quickly" oxidized.
*We have removed the word quickly.*

p2, L10. Give reference for the direct aerosol effect comment
215 *We have added a reference to (Charlson et al., 1990).*

p2, L11. Add radiation as one of the major drivers of BVOC emissions
*We have added radiation as one of the main drivers.*

220 p2, L12. It is usually good to cite articles if possible, and could have used e.g. Hantson et al 2017 or Schurgers et al. 2009 here.
*We added the Hantson et al. (2017) reference.*

p2, L13. There are several studies suggesting that increased CO2 can inhibit BVOC emissions (e.g. Arneth et al, 2007, refs
225 in Hantson et al 2017). This is also a major source of uncertainty that needs a mention.
*We have added this sentence: "There are however a range of uncertainties associated with these feedbacks, including $CO_2$ inhibition of BVOC emissions (Arneth et al., 2007)."*

p2, L22. Season matters. BSOA often dominates SOA in summertime, but there is plenty of evidence that in wintertime
230 wood-burning often dominates or plays a major role (e.g. Brown et al., 2016, Glasius et al., 2018).
*A sentence has been added on regional and seasonal differences and we have included a reference to (Glasius et al., 2018). The new sentence reads: "However, there are large seasonal and regional differences in sources, with BSOA dominating in sum-*

*mer, while sources like wood burning can be more dominant in winter, in particular in populated regions (Glasius et al., 2018)."*

235    p2, L24. Although the Spracklen et al 2011 study was very innovative and interesting, there are several issues with the conclusions, see e.g. Hodzic  Jimenez, 2011. This question is very complex and unresolved as far as I know.

*We have reworded this sentence somewhat and it now reads: "Moreover, some studies have found that the biogenic SOA formation is anthropogenically controlled (Spracklen et al., 2011; Kanakidou et al., 2000; Carlton et al., 2010)."*

240    p3, L30-33. I think the sentences starting on L30 are very clear, fair, and with the important caveat represented by the last sentence. These lines could usefully be used in the abstract.

*We have added the following sentence to the abstract: "The goal of the study is to investigate whether it is of importance to treat SOA formation processes correctly in the models, rather than to evaluate the correctness of the current treatment in the models."*

245    p5, L2. The descriptions are brief, not "detailed".

*We have added "more" before detailed to highlight that the descriptions in the next sections are more detailed than the brief description in this paragraph.*

       p5, L13. Which MEGAN version?

250 *We have included "version 2.1" to the text to clarify this.*

       p6, L1. So, is this a new version of NorESM, or a version used just for this study? If the latter, then the conclusions aren't relevant to other NorESM work, which would seem to remove some of the point of including this model.

*This is not a new version of the model, but a minor modification of it. This modification is included in future versions of the*

255 *model so the results from this study are still relevant.*

       p6, L9. I think you mean hydroxyl, not hydroxide, and better to say nitrate radical, as nitrate is often used for the aerosol compound.

*We have changed this according to the reviewer's suggestion.*

260

       p6, L11. This was confusing. If I understand right, one has two types of ELVOC then, one that can influence NPF, and the other behaves exactly as L/SVOC. As you ELVOC and L/SVOC compounds have the same mass, why not simply put the non-NPF ELVOC in as L/SVOC?

*The idea behind type of parametrizing of the ELVOCs in NorESM is that not all VOC formed from the oxidation of monoterpene*

265 *by ozone have low enough volatility to contribute to new particle formation (NPF). Therefore, only 50 %. of the ELVOCs can participate in NPF. The advantage of this treatment is that we essentially have three types of VOCs in NorESM but only two*

*tracers. This is not used in the version of the model used in this paper but has been utilized in new developments of the model.*

p7, L32. This was also confusing. Table 1 suggests that ECHAM uses fixed yields of L/SVOC and ELVOC, but L32 suggests partitioning depends on pre-existing organic mass. And what is meant by pre-existing OM? Does this influence the DRE/CRE calculations?

*The yields are fixed. The partitioning concerns the partitioning to the different aerosol modes. We have changed the text to clarify this and the text now reads: "The SOA formation mechanism (Jokinen et al., 2015) includes both kinetic condensation to Fuchs-corrected surface area (condensation sink). Moreover, the relative partitioning to the Aitken/accumulation/coarse mode is done according to pre-existing organic mass in the respective modes."*

*All organic mass is considered in the DRE/CRE calculations.*

p9, L2-7. I am not sure the argument about interactive oxidants can explain a factor of 3. Sure, when the BVOC are emitted one can expect reduced OH and other oxidants, but this just delays the oxidation close to the surface. Isoprene which isn't oxidized near the surface will still be oxidized a little further up in the troposphere. Why would the total amount change by a factor of 3? Did you check changes in oxidant fields associated with this argument?

*We have written in the article that this can partly be explained by the interactive oxidants and thus do not claim that the inter-active oxidants can explain a factor of 3. If isoprene is oxidised further away from the sources it will increase the lifetime of the isoprene which also increases the column burden. Not only the average level of the oxidants but also the diurnal variation of the both BVOCs and oxidants and their timing will matter for the lifetime and of the BVOCs and affect how close to the sources they are oxidised. We have not compared the oxidant fields between the models since we don't have this output from all the models.*

p16-17. This is where I think the limitations and results of this study need to be put alongside the many other uncertainties surrounding SOA modelling.

*We have added a paragraph on limitations and uncertainties regarding the SOA parameterizaitons in the ESMs used in this study here.*

p26, Fig. 1. State which years are shown here. Also, I was surprised to see no errorbars on the ECHAM runs, and that even those for NorESM were so small. Did ECHAM also just use one fixed year (2000) of BVOC emission, same as EC-Earth?

*Yes, ECHAM uses the emissions from one fixed year. We have added information regarding this to the figure text.*

**References**

[revised manuscript text omitted]